# Elucidating Softening Mechanism of Honey Peach (*Prunus persica* L.) Stored at Ambient Temperature Using Untargeted Metabolomics Based on Liquid Chromatography-Mass Spectrometry

**Xiaoxue Kong** [1,2,†]**, Haibo Luo** [2,†]**, Yanan Chen** [2]**, Hui Shen** [1]**, Pingping Shi** [1]**, Fang Yang** [1]**, Hong Li** [3,*] **and Lijuan Yu** [1,*]

[1] Agro-Products Processing Research Institute, Yunnan Academy of Agricultural Sciences, Kunming 650221, China; 45225@njnu.edu.cn (X.K.); shenhui@yaas.org.cn (H.S.); spp@yaas.org.cn (P.S.); yf@yaas.org.cn (F.Y.)

[2] School of Food Science and Pharmaceutical Engineering, Nanjing Normal University, Nanjing 210023, China; 45235@njnu.edu.cn (H.L.); chenyn813@163.com (Y.C.)

[3] College of Food Science and Technology, Yunnan Agricultural University, Kunming 650201, China

[*] Correspondence: lihong@yaas.org.cn (H.L.); ylj@yaas.org.cn (L.Y.)

[†] These authors contributed equally to this work.

**Abstract:** Peach fruit softening is the result of a series of complex physiological and biochemical reactions that influence shelf life and consumer acceptance; however, the precise mechanisms underlying softening remain unclear. We conducted a metabolomic study of the flesh and peel of the honey peach (*Prunus persica* L.) to identify critical metabolites before and after fruit softening. Compared to the pre-softening profiles, 155 peel metabolites and 91 flesh metabolites exhibited significant changes after softening ($|\log_2(FC)| > 1$; $p < 0.05$). These metabolites were mainly associated with carbohydrate metabolism, respiratory chain and energy metabolism (citrate cycle, pantothenate and CoA biosynthesis, nicotinate and nicotinamide metabolism, and pentose and glucuronate interconversions), reactive oxygen species (ROS) metabolism, amino acid metabolism, and pyrimidine metabolism. During peach fruit softening, energy supply, carbohydrate and amino acid metabolism, oxidative damage, and plant hormone metabolism were enhanced, whereas amino acid biosynthesis and cell growth declined. These findings contribute to our understanding of the complex mechanisms of postharvest fruit softening, and may assist breeding programs in improving peach fruit quality during storage.

**Keywords:** honey peach; softening; untargeted metabolomics; LC–MS; metabolites

## 1. Introduction

Honey peach (*Prunus persica* L.; family *Rosaceae*) fruit is an important horticultural product cultured worldwide for its pleasant aroma, juicy texture, delicate flavor, and rich nutrient content [1]. Honey peaches are rich in phytochemicals, including lipids, vitamins, nucleotides, phenolics (phenolic acids and flavonoids), carotenoids, triterpenes, and alkaloids [2]. Many phytochemicals possess health-promoting benefits such as free radical neutralization, cancer prevention, and heart disease prevention [3]. However, honey peaches are climacteric fruits with a vigorous postharvest respiratory physiological metabolism. Honey peach softening refers to the transition of the fruit from a ripe stage to an overripe stage, where moderate softening is a sign of complete maturity. Many phytochemicals are formed during the softening process [4], although excessive softening leads to postharvest quality deterioration, storage and transportation limitations, and reduced shelf life and market value.

Fruit softening involves a series of complex physiological and metabolic processes. Fruit softening during storage is generally thought to be caused mainly by cell wall





structural alteration and degradation. Pectin, cellulose, hemicellulose, and other plant polysaccharides are the main components of most plant cell walls and play key roles in maintaining cell structure [5,6]. Comparative proteomics analysis of peaches at different ripening stages revealed that the differentially expressed proteins were mainly involved in cellular activities such as sugar metabolism, membrane structure, and cell-cycle control; in particular, polygalacturonase, pectate lyase, calmodulin, and calcineurin B-like protein exhibited functional roles in controlling fruit development and maintaining textural integrity during ripening [7–10]. In addition, several studies have found that plant hormone regulation, starch degradation, and energy metabolism are involved in fruit softening. Specifically, ethylene and abscisic acid play important regulatory roles in the final stage of peach ripening. Treatment with exogenous ethylene, which regulates respiration in climacteric fruit such as peaches, rapidly reduced fruit hardness, whereas 1-MCP treatment significantly delayed softening [11,12]. Abscisic acid is an important regulatory factor of fruit senescence after ripening, speeding up ripening and softening processes [13]. Amylase-catalyzed starch degradation increased the contents of soluble solids and reduced sugars, resulting in decreased fruit firmness [14,15]; therefore, postharvest starch degradation and sucrose metabolism may also contribute to peach softening. However, peach softening is a complex process, and its precise phytochemical variations and metabolic mechanism remain to be clarified.

Metabolomics is a powerful strategy for effectively identifying and quantifying metabolites within cells or tissues [16,17], providing an impartial approach for investigating correlations among interconnected metabolites via multiple pathways [18]. In recent years, metabolomics has been used to investigate the metabolic mechanisms underlying peach ripening and senescence [19]. The most commonly employed analytical techniques are liquid chromatography (LC)–tandem mass spectrometry (MS/MS) and nuclear magnetic resonance (NMR). Compared to NMR, LC-MS/MS offers superior resolution of chromatographic peaks, heightened sensitivity, and greater efficiency [20,21]. Untargeted metabolomics, a widely employed approach for qualitative sample analysis, can rapidly identify and classify metabolites based on differences in metabolic pathway maps, and based on LC-MS/MS, can reliably analyze metabolic profiles [22,23].

The objective of this study was to elucidate the softening mechanism of postharvest peaches. We performed global untargeted metabolomic profiling via LC-MS to study the mechanistic variation in peaches harvested at 90% maturity (pre-softening) and stored for 4 days at 25 ± 1 °C and 80–90% relative humidity (post-softening). We identified differential metabolites and analyzed the associated metabolic pathways. Our findings clarify the mechanism underlying peach softening, and support metabolic regulation to extend their shelf life, thereby reducing peach storage and transportation losses.

## 2. Materials and Methods

### 2.1. Analytical Standards and Reagents

Analytical standards were purchased from Thermo Fisher Scientific (Waltham, MA, USA), including methanol (≥99%; CAS no.: 67-56-1), acetonitrile (≥99%; CAS no.: 75-05-9), and formic acid (LC-MS grade; CAS no.: 64-18-6). The major reagents were purchased from Nanjing Jiancheng Bioengineering Institute (Nanjing, China), including dihydrogen phosphate potassium (≥99%; CAS no.: 7778-77-0), dipotassium hydrogen phosphate (≥99%; CAS no.: 7758-11-4), and L-2-chlorobenzalanine (≥98.5%; CAS no.: 103616-89-3).

### 2.2. Plant Materials and Treatments

Fresh peaches were hand-harvested from a *Prunus persica* L. orchard in Laishan, Shandong Province, China. All samples were similar in size and color, and lacking visible defects. To investigate the softening mechanism, samples were stored at 25 ± 1 °C and relative humidity of 80–90% for 4 days; hard peaches from the day of harvest (day 0) were used as the control.

The peel and flesh of hard peaches (PHP and FHP, respectively) and stored peaches (PSP and FSP, respectively) were sampled using a sharp stainless steel knife, cut into small pieces (3–5 mm$^3$), frozen with liquid nitrogen, and stored at −80 °C until analysis.

### 2.3. Visualization of the Ultrastructure

The cell ultrastructure of peach peel and flesh were visualized as previously described by Luo et al. (2019), with some modifications [24]. Tissue blocks of approximately 1 mm$^3$ were sliced from peach surface and washed three times with cold phosphate-buffered saline (PBS, pH7.0, 0.1 M) for 15 min each. The samples were soaked in 2.5% (*w/v*) glutaraldehyde for 24 h at 4 °C, washed with PBS three times, and then soaked in 1% osmic acid fixative solution for 2 h. The samples were washed with PBS (pH7.4) three times, and dehydrated in 50%, 70%, and 90% ethanol for 15 min each, followed by 100% ethanol for 20 min. After fixing with conductive carbon adhesive and spray gold with an ion sputtering instrument for 50 s, and the slices were observed under a FEI Nova Nano 450 scanning electron microscope (FEI Company, Hillsboro, OR, USA).

### 2.4. Sample Preparation for LC-MS

For each sample, 80 mg was transferred to a 1.5-mL Eppendorf tube containing two small steel balls. Then, 1 mL of a methanol and water mixture (7:3, *v/v*) was added and the tube was placed in a −20 °C freezer for 2 min. Next, the sample was ground at 60 Hz for 2 min, vortexed, and ultrasonicated at ambient temperature for 30 min. The tube was then stored at −20 °C for 12 h before centrifugation for 10 min (10,000× *g*, 4 °C). From each sample tube, 150 μL of supernatant was filtered through a 0.22 μm organic-phase pinhole filter and transferred to an LC vial, which was stored at −80 °C until LC-MS analysis.

To avoid instrument errors, quality control (QC) samples were prepared by mixing all samples in equal volumes and analyzed to test the stability of the instrument system and the repeatability of sampling.

### 2.5. Ultra-High-Performance LC with Quadrupole Time-of-Flight Mass Spectrometry (UPLC-Q-TOF-MS)

UPLC-Q-TOF-MS analysis was performed using a Nexera UHPLC (Shimadzu, Kyoto, Japan) combined with a Q-Exactive high-resolution MS (Thermo Fisher Scientific). Samples were separated with an ACQUITY UPLC HSS T3 column (100 mm × 2.1 mm, 1.8 μm; Waters Corp., Milford, MA, USA) following the manufacturer's procedure. The binary gradient elution system consisted of (A) water containing 0.1% formic acid and (B) acetonitrile containing 0.1% formic acid. The injection volume was 2 μL, the column temperature was 45 °C, and the flow rate was 0.35 mL min$^{-1}$. The separation gradient was as follows: 0 min, 5% B; 4 min, 30% B; 8 min, 50% B; 10 min, 80% B; 14 min, 100% B; 15.1 min, 5% B; and 16 min, 5% B.

Mass spectrometric data were acquired with a Q-Exactive Plus MS (Thermo Fisher Scientific, Waltham, MA, USA) with an electrospray ionization source. The MS parameters were as follows: source spray voltage of 3.00 kV in the negative and 3.50 kV in the positive ion mode, and capillary temperature of 320 °C. All data were collected in MS$^E$ mode, with a scan range of 100–1200, a full scan at a resolution of 70,000, and a normalized collision energy of 30 eV. Data were collected in data-dependent acquisition or MS/MS mode again to obtain more fragment ions and detailed information pertaining to metabolites.

### 2.6. Metabolome Data Analysis

The Progenesis QI v2.3 software (Nonlinear Dynamics, Newcastle, UK) was employed for baseline filtering, retention time correction, peak identification and alignment, and peak area normalization. The main parameters were a precursor tolerance of 5 mg L$^{-1}$, product tolerance of 10 mg L$^{-1}$, and production threshold of 5%. Compounds were identified based on their mass-to-charge ratio (*m/z*), secondary fragments, and isotopic distribution using the plant metabolome database. Each analysis was performed six times and pre-processed

by subtracting the blank response and aligning according to the QC sample. Ion peaks with all missing values (0 value) > 50% in the group were deleted. Compounds obtained qualitatively were screened according to their qualitative result scores; those with scores below 36 (out of 60) were regarded as inaccurate and deleted.

For multivariate statistical analysis, normalized data were imported into SIMCA-P v13.0 (Umetrics AB, Umea, Sweden). The processed data were analyzed using principal component analysis (PCA) to observe the overall distribution among the samples and the stability of the whole analysis methodology. Orthogonal partial least-squares discriminant analysis (OPLS-DA) was used to distinguish metabolites that differed between the pre- and post-softening groups. To prevent overfitting, seven-fold cross-validation and 200-response permutation testing were performed to evaluate model quality. Univariate statistics mainly included Student's *t*-test and fold change (FC) analysis to compare metabolites between two groups. Differential metabolites between the pre- and post-softening groups were selected based on a variable importance of projection (VIP) score > 1, $p < 0.05$, and m (i.e., $|\log_2(FC)| > 1$) [25].

Differential metabolites identified using LC-MS and associated with diverse pathways were visualized by plotting a heatmap (http://www.r-project.org, accessed on 26 May 2023) and analyzed via metabolomics pathway analysis (http://www.metaboanalyst.ca/, accessed on 27 May 2023). The Kyoto Encyclopedia of Genes and Genomes (KEGG) database (http://www.kegg.jp/, accessed on 27 May 2023) was used to determine the position and function of each metabolite in various metabolic pathways.

## 3. Results

### 3.1. Cellular Ultrastructure of Peaches before and after Softening

The morphologies of peach peel and flesh before softening (day 0; the day of harvesting) and after softening (day 4 of storage at 25 °C) were observed using scanning electron microscopy. Before softening, the fruit cells were compact, full, uniform in size, and closely arranged, and the cell edges were clearly visible (Figure 1A). After softening, the intercellular space increased, the edges of some cells became obscured with no evident boundary, and there were different degrees of contractions and folds, indicating that the cell structure of the fruit was damaged to an extent (Figure 1B).

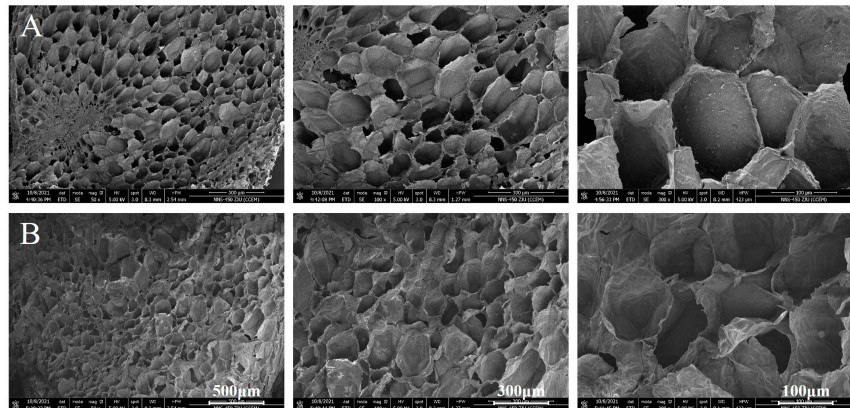

**Figure 1.** The cell ultrastructure of peach fruit before (**A**) and after (**B**) softening.

### 3.2. Metabolite Identification

The five sample groups (QC, PSP, FSP, PHP, and FHP) were analyzed using UPLC-Q-TOF-MS. In total, 7778 and 5577 precursor molecules were extracted in positive and negative ion modes, respectively. Progenesis QI v2.3 software was applied to process the raw UPLC-Q-TOF-MS data. Ultimately, 1660 metabolite ion features were detected. Detailed information regarding the metabolites, including pathway analysis, chemical analysis, *m/z* values, retention time, exact mass, molecular formula, mass error, precursor type, CAS number, and KEGG code, are presented in Table S1.

### 3.3. QC and Identification of Differential Metabolites

PCA, an unsupervised multivariate analysis, was performed to evaluate the stability of the system. In the score plots in Figure 2A, which were obtained from seven-fold cross-validation, the QC samples were clustered together, indicating satisfactory stability and reproducibility of the UPLC-Q-TOF-MS method. The six replicates of each group were clearly separated. The first two PCs explained 59.6% and 20.2% of the total variance, respectively. To more intuitively display the relationship between the QC samples and other samples, we conducted hierarchical clustering of the expression levels of all metabolites (Figure 2B).

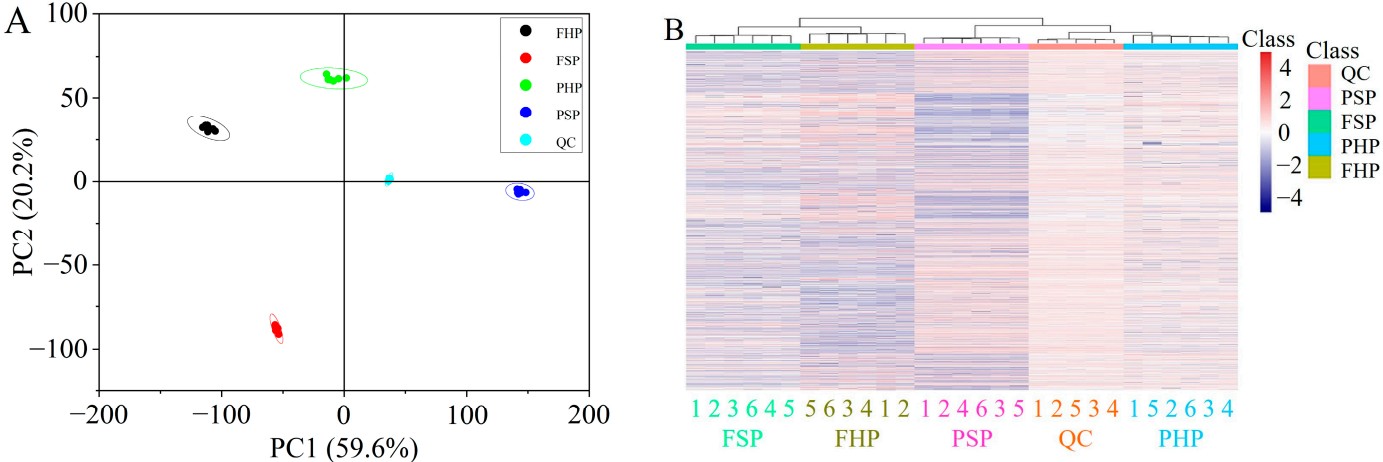

**Figure 2.** PCA score chart (**A**) and heatmap (**B**) of all samples.

To further confirm the differential metabolites between pre- and post-softening of peach peel (PSP/PHP) and flesh (FSP/FHP) samples, and filter out irrelevant components, OPLS-DA was used to maximize the differences between the groups PSP/PHP and FSP/FHP (Figure 3). Parameter values ($R^2X$, $R^2Y$, and $Q^2$) closer to 1 indicated a more stable and reliable model; the values for the PSP/PHP and FSP/FHP models were 0.967 and 0.965 for $R^2X$, 1 and 0.999 for $Q^2$, and 1 and 1 for $R^2Y$. These results indicated that the mathematical models showed high predictive accuracy, and could be used to identify differential metabolites.

The following criteria were applied to identify significantly differential metabolites using the criteria VIP > 1, $p < 0.05$, and $|\log_2(FC)| > 1$. In total, 155 metabolites were selected in the groups PSP/PHP (81 upregulated, 74 downregulated), and 93 metabolites were selected in the groups FSP/FHP (50 upregulated, 43 downregulated). The numbers of differential metabolites are shown in Figure 4. Differential metabolites were visualized using volcano plots, with red and blue dots representing significantly up- and downregulated metabolites, respectively, and gray dots representing metabolites without significant changes (Figure 4). During the peach softening process, there were significant differences in metabolites in both peel and flesh, with only a few metabolites remaining unchanged. The identified metabolites were classified into 11 super-classes according to their KEGG annotations. The distribution is shown in Figures 5 and S1, and the differential metabolites in peaches before and after softening are listed in Table 1.

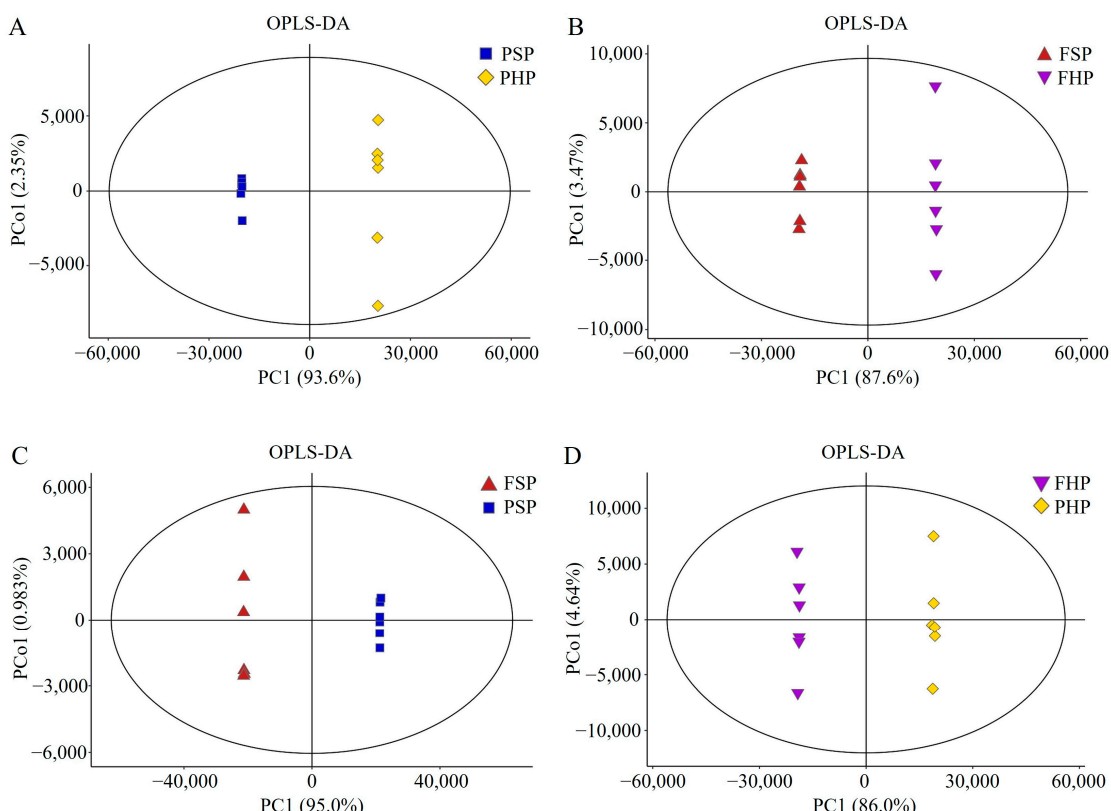

**Figure 3.** OPLS-DA score chart of PSP + PHP (**A**) and FSP + FHP (**B**). (**C**) FSP + PSP and (**D**) FHP + PHP.

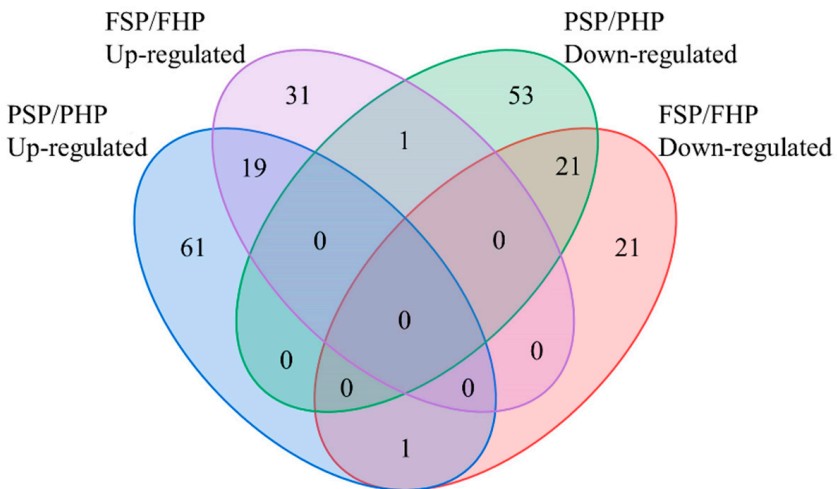

**Figure 4.** Number of differential metabolites in peach fruit before and after softening.

*3.4. Hierarchical Clustering Analysis (HCA)*

To directly evaluate differences in metabolite expression between the groups, we conducted HCA of the top 40 differential metabolites (Figure 6). Most were lipids and lipid-like molecules. In PSP/PHP, lipid-like molecules accounted for 35.9% of differential metabolites, and included the upregulated priverogenin B ($|\log_2(FC)|$: 37.26), goyaglycoside f (15.04), and lucidumol A (8.92) and the downregulated pitheduloside B (5.08), zedoarol (3.12), and angelic acid (2.02). In FSP/FHP, lipid-like molecules accounted for 34.04% of differential metabolites, and included the upregulated 10'-apo-beta-caroten-10'-al ($|\log_2(FC)|$: 35.19), corchorifatty acid F (4.50), and tragopogonsaponin B (3.77) and the downregulated goshon-

oside F3 (35.34), 3-O-cis-coumaroylmaslinic acid (4.49), and deoxynivalenol 3-glucoside (3.90). In addition, orotidine content was upregulated in both PSP/PHP ($|\log_2(FC)|$: 36.14) and FSP/FHP (36.16). Glutathione (GSH; $|\log_2(FC)|$: 37.25), uridine diphosphate-D-xylose (UDP-D-xylose; 35.86), N-gamma-L-glutamyl-D-alanine (35.16), procyanidin B1 (9.52), and procyanidin B2 (8.67) increased significantly only in FSP/FHP. Overall, these differential metabolites were related to changes in cell membrane lipid oxidation, energy production, pectin biosynthesis, characteristic volatile components, and color.

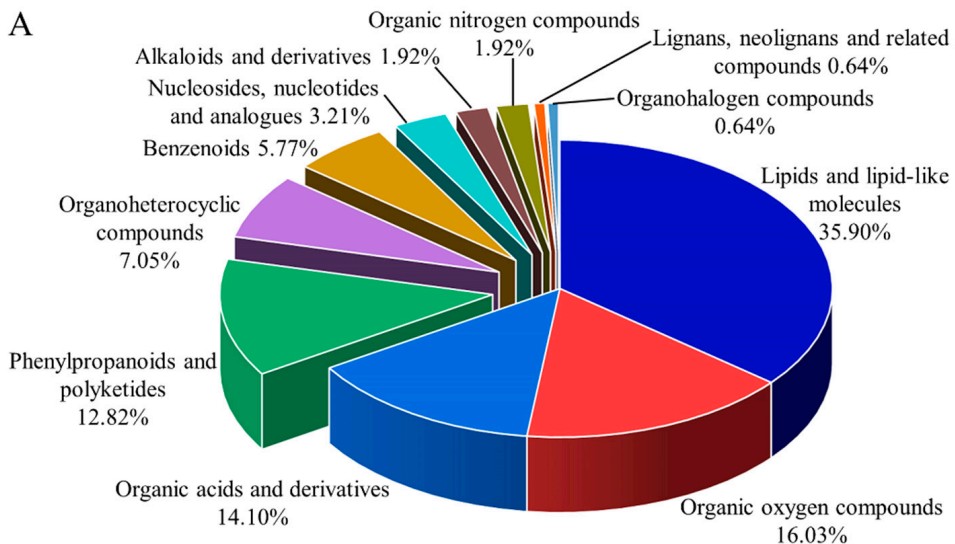

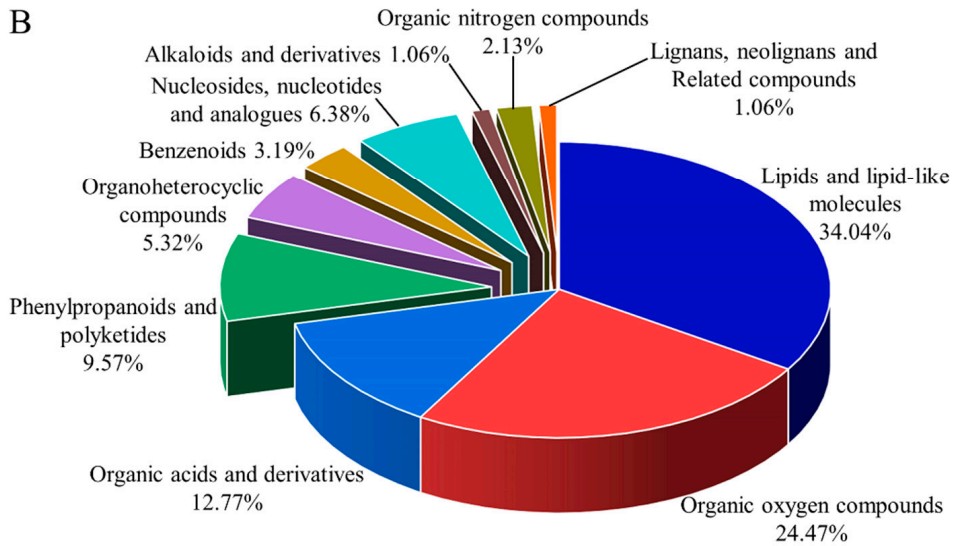

**Figure 5.** The super class distribution of identified differential metabolites covering 11 groups categorized according to their molecular structure. (**A**) PSP/PHP; (**B**) FSP/FHP.

**Table 1.** The differential metabolites in peach fruit before and after softening.

| No. | ID | *m/z* | Retention Time (min) | Ion Mode | Metabolites | Compound ID | PSP/PHP | FSP/FHP |
|---|---|---|---|---|---|---|---|---|
| | Alkaloids and derivatives | | | | | | | |
| 1 | 5.03_553.2138m/z | 553.2138 | 5.0349167 | neg | Dehydroaporheine | HMDB0033355 | 5.5528348 | |
| 2 | 0.91_675.0976m/z | 675.09764 | 0.9134167 | neg | Prebetanin | HMDB0029411 | −1.1931527 | −0.2097159 |
| 3 | 0.82_137.0476n | 160.0368 | 0.8212167 | pos | Trigonelline | HMDB0000875 | −6.296828 | −3.4098136 |

**Table 1.** *Cont.*

| No. | ID | *m/z* | Retention Time (min) | Ion Mode | Metabolites | Compound ID | PSP/PHP | FSP/FHP |
|---|---|---|---|---|---|---|---|---|
| Benzenoids | | | | | | | | |
| 4 | 12.05_333.1354m/z | 333.13543 | 12.045667 | neg | 4′-Methoxymucidin | HMDB0030019 | 3.3352022 | |
| 5 | 12.59_501.2238m/z | 501.2238 | 12.591167 | neg | Purothionin AII | HMDB0039001 | 2.8072538 | |
| 6 | 10.98_292.2037n | 293.21085 | 10.978267 | pos | [7]-Paradol | HMDB0040806 | 2.5051899 | |
| 7 | 5.16_437.2030m/z | 437.20305 | 5.15925 | neg | N-Phenyl-2-naphthylamine | HMDB0032865 | 2.0844425 | 2.2331908 |
| 8 | 5.03_524.1345m/z | 524.13452 | 5.0319 | pos | Protohypericin | HMDB0034180 | −1.2897264 | |
| 9 | 15.27_150.1277m/z | 150.12768 | 15.2738 | pos | p-Mentha-1,3,5,8-tetraene | HMDB0029641 | −1.422287 | −0.6103573 |
| 10 | 0.91_521.1087m/z | 521.10873 | 0.9064 | pos | Isomelitric acid A | HMDB0039523 | −1.580731 | −0.6180271 |
| 11 | 4.72_493.1289m/z | 493.12889 | 4.7210667 | pos | Palmidin A | HMDB0034038 | −1.5901031 | −1.6091476 |
| 12 | 1.83_278.1516n | 301.14084 | 1.83035 | pos | Dibutyl phthalate | HMDB0033244 | −1.5974966 | −0.7392945 |
| 13 | 4.72_583.1255m/z | 583.12551 | 4.7247833 | neg | Rheidin C | HMDB0038508 | | −1.2387168 |
| Lignans, neolignans, and related compounds | | | | | | | | |
| 14 | 5.39_522.2105n | 567.20854 | 5.3948167 | neg | Isolariciresinol 4′-O-beta-D-glucoside | HMDB0040471 | 1.220256 | |
| 15 | 5.18_567.2084m/z | 567.20838 | 5.1766 | neg | Isolariciresinol 9-O-beta-D-glucoside | HMDB0032907 | 0.793055 | −1.1147439 |
| Lipids and lipid-like molecules | | | | | | | | |
| 16 | 15.26_474.3706n | 497.35981 | 15.256067 | pos | Priverogenin B | HMDB0034644 | 37.261839 | |
| 17 | 12.40_781.4695m/z | 781.46946 | 12.4023 | pos | Goyaglycoside f | HMDB0037124 | 15.036405 | |
| 18 | 15.10_472.3550n | 495.34426 | 15.096133 | pos | Lucidumol A | HMDB0033233 | 8.9189368 | |
| 19 | 14.51_446.3394n | 469.32866 | 14.510817 | pos | Secasterone | HMDB0040999 | 5.2741667 | |
| 20 | 15.13_448.3551n | 471.3443 | 15.131617 | pos | 6-Deoxodolichosterone | HMDB0034332 | 5.1783877 | |
| 21 | 14.07_643.4173m/z | 643.41728 | 14.06755 | pos | Fasciculic acid A | HMDB0036439 | 3.6356945 | |
| 22 | 13.89_585.3757m/z | 585.3757 | 13.894517 | pos | Ganoderic acid Mg | HMDB0035999 | 2.9360617 | |
| 23 | 5.31_192.1514n | 175.14811 | 5.3125 | pos | gamma-Ionone | HMDB0034979 | 2.7246443 | 2.3180287 |
| 24 | 8.73_518.3244n | 563.32257 | 8.7302167 | neg | Ganolucidic acid C | HMDB0039691 | 2.462305 | |
| 25 | 14.21_508.3764n | 531.36563 | 14.208733 | pos | Fasciculol C | HMDB0035853 | 2.2688443 | |
| 26 | 9.66_535.2879m/z | 535.28789 | 9.6555167 | pos | Corchoroside A | HMDB0033846 | 2.0353167 | |
| 27 | 11.96_633.3968m/z | 633.39684 | 11.955867 | pos | Calenduloside E | HMDB0040851 | 1.9493374 | |
| 28 | 8.28_518.3234n | 563.32252 | 8.2771333 | neg | Ganoderic acid C2 | HMDB0035304 | 1.9198447 | |
| 29 | 5.16_415.1936m/z | 415.19356 | 5.1589 | pos | S-Furanopetasitin | HMDB0036131 | 1.852255 | 1.1850532 |
| 30 | 5.50_415.1975m/z | 415.1975 | 5.5021833 | neg | (3S,7E,9S)-9-Hydroxy-4,7-megastigmadien-3-one 9-glucoside | HMDB0036822 | 1.7683917 | 1.4457104 |
| 31 | 5.29_377.1817m/z | 377.18167 | 5.2873333 | neg | 6Z-8-Hydroxygeraniol 8-O-glucoside | HMDB0035025 | 1.7644214 | 1.3697986 |
| 32 | 5.58_373.1868m/z | 373.1868 | 5.5755667 | neg | 6-Epi-7-isocucurbic acid glucoside | HMDB0029782 | 1.755631 | |
| 33 | 4.74_379.1610m/z | 379.161 | 4.7429333 | neg | Prenyl arabinosyl-(1->6)-glucoside | HMDB0041360 | 1.6820333 | |
| 34 | 5.47_282.1467n | 281.13938 | 5.4670833 | neg | Epidihydrophaseic acid | HMDB0038661 | 1.6585013 | 2.5180264 |
| 35 | 11.35_294.2193n | 295.22654 | 11.354317 | pos | 2-Hydroxylinolenic acid | HMDB0031103 | 1.6570269 | −1.0014807 |
| 36 | 9.15_502.3297n | 547.32789 | 9.1458 | neg | Ganolucidic acid B | HMDB0035751 | 1.6085416 | |
| 37 | 5.41_441.1978m/z | 441.1978 | 5.4130167 | neg | 1-Hexanol arabinosylglucoside | HMDB0031689 | 1.6013305 | |
| 38 | 5.36_471.1872m/z | 471.18718 | 5.35895 | neg | 11,13-Dihydrotaraxinic acid glucosyl ester | HMDB0035867 | 1.5551523 | 1.0746529 |
| 39 | 8.75_500.3135n | 483.3102 | 8.7520667 | pos | Ganolucidic acid A | HMDB0035302 | 1.5458326 | |
| 40 | 7.71_695.4014m/z | 695.40141 | 7.7110833 | neg | Momordicoside E | HMDB0035697 | 1.531432 | |
| 41 | 5.36_433.2079m/z | 433.20795 | 5.35895 | neg | Dihydroroseoside | HMDB0040614 | 1.4165233 | |
| 42 | 9.55_502.3292n | 503.33642 | 9.5534167 | pos | Medicagenic acid | HMDB0034551 | 1.3474913 | |
| 43 | 5.39_194.1670n | 177.16372 | 5.3880333 | pos | 5-Isopropyl-2-(2-methylpropyl)-2-cyclohexen-1-one | HMDB0038216 | 1.3416272 | |
| 44 | 9.57_410.3181n | 433.30983 | 9.5741 | pos | (6alpha,22E)-6-Hydroxy-4,7,22-ergostatrien-3-one | HMDB0037380 | 1.3000743 | |
| 45 | 5.02_433.2080m/z | 433.20803 | 5.0172833 | neg | 9,13-Dihydroxy-4-megastigmen-3-one 9-glucoside | HMDB0036318 | 1.2650463 | |
| 46 | 5.34_393.1768m/z | 393.17677 | 5.3405333 | neg | Nepetariaside | HMDB0039014 | 1.2443563 | 0.5806051 |

**Table 1.** *Cont.*

| No. | ID | *m/z* | Retention Time (min) | Ion Mode | Metabolites | Compound ID | PSP/PHP | FSP/FHP |
|---|---|---|---|---|---|---|---|---|
| 47 | 4.12_451.2187m/z | 451.21868 | 4.12275 | neg | Kiwiionoside | HMDB0038691 | 1.2365472 | |
| 48 | 5.09_427.1938m/z | 427.19381 | 5.0859667 | pos | Pisumionoside | HMDB0039947 | 1.2300592 | |
| 49 | 4.91_282.1466n | 281.13946 | 4.9062833 | neg | Pisumic acid | HMDB0039241 | 1.1927666 | 2.1976093 |
| 50 | 5.29_332.1832n | 355.17243 | 5.2941167 | pos | (2E,4E,7R)-2,7-Dimethyl-2,4-octadiene-1,8-diol 8-O-b-D-glucopyranoside | HMDB0038747 | 1.16536 | 0.727943 |
| 51 | 5.19_439.1822m/z | 439.18219 | 5.1942167 | neg | cis-3-Hexenyl b-primeveroside | HMDB0031690 | 1.1648099 | |
| 52 | 4.85_386.1940n | 431.19223 | 4.8532667 | neg | Citroside A | HMDB0030370 | 1.1464168 | 0.5008677 |
| 53 | 9.57_468.3237n | 469.33097 | 9.5741 | pos | Uralenolide | HMDB0038797 | 1.1359664 | |
| 54 | 9.15_502.3291n | 503.33641 | 9.14545 | pos | Esculentic acid (Phytolacca) | HMDB0034639 | 1.119195 | |
| 55 | 5.03_348.1781n | 371.16731 | 5.0319 | pos | Foeniculoside V | HMDB0034874 | 1.1036398 | 2.628105 |
| 56 | 5.95_421.2081m/z | 421.20813 | 5.94505 | neg | 1-Octen-3-yl primeveroside | HMDB0032960 | 1.0953733 | 2.8884386 |
| 57 | 4.80_433.2080m/z | 433.20798 | 4.7979 | neg | Icariside B8 | HMDB0036846 | 1.0514936 | |
| 58 | 5.48_280.1311n | 279.1238 | 5.4847667 | neg | Nigellic acid | HMDB0036094 | 1.0289409 | 1.9319225 |
| 59 | 10.99_679.3853m/z | 679.38531 | 10.9895 | neg | 2alpha-Hydroxypyracrenic acid | HMDB0029780 | 1.0173618 | |
| 60 | 4.91_264.1360n | 265.14329 | 4.90995 | pos | 3-Epiarmefolin | HMDB0036135 | 0.6841163 | 1.4082933 |
| 61 | 11.35_454.3443n | 455.35163 | 11.354317 | pos | Ursonic acid | HMDB0036007 | −0.4947675 | −1.8832459 |
| 62 | 5.57_458.1786n | 481.16784 | 5.56605 | pos | Deoxynivalenol 3-glucoside | HMDB0039852 | −0.5041028 | −3.9033631 |
| 63 | 11.34_473.3624m/z | 473.36241 | 11.336667 | pos | 27-Hydroxyisomangiferolic acid | HMDB0036064 | −0.6364165 | −1.9997725 |
| 64 | 0.81_344.1316n | 389.12984 | 0.81115 | neg | Lactitol | HMDB0040937 | −0.8733643 | −1.2932778 |
| 65 | 12.98_438.3496n | 439.35686 | 12.97695 | pos | Thujyl 19-trachylobanoate | HMDB0036840 | −0.9976602 | −2.3387444 |
| 66 | 0.79_207.0503m/z | 207.05031 | 0.7941333 | neg | 3-Hydroxymethylglutaric acid | HMDB0000355 | −1.032756 | −0.6809323 |
| 67 | 2.14_346.1261n | 369.11535 | 2.1447 | pos | Aucubin | HMDB0036562 | −1.057286 | −2.7078305 |
| 68 | 12.99_457.3672m/z | 457.36724 | 12.99425 | pos | beta-Elemolic acid | HMDB0034961 | −1.2980053 | −3.2925127 |
| 69 | 13.01_410.3545n | 411.36175 | 13.011333 | pos | Delta 8,14 -Sterol | HMDB0006928 | −1.3317368 | −2.134609 |
| 70 | 6.98_292.1883n | 315.17763 | 6.9816333 | pos | (S)-3-Octanol glucoside | HMDB0032958 | −1.3828306 | −0.5888407 |
| 71 | 11.39_277.1797m/z | 277.1797 | 11.38935 | pos | Phytuberin | HMDB0035754 | −1.4766195 | −0.5807962 |
| 72 | 14.14_310.3102m/z | 310.31019 | 14.137933 | pos | Geranylcitronellol | HMDB0032147 | −1.5039805 | |
| 73 | 1.13_118.0865m/z | 118.08646 | 1.1279 | pos | Angelic acid | HMDB0029608 | −2.0222781 | −0.4844698 |
| 74 | 7.07_414.2252n | 437.21441 | 7.0656833 | pos | (4R,5S,7R,11S)-11,12-Dihydroxy-1(10)-spirovetiven-2-one 11-glucoside | HMDB0033150 | −2.3741167 | −0.7120234 |
| 75 | 6.47_264.1362n | 263.12892 | 6.4695167 | neg | Alkhanin | HMDB0036202 | −2.8303188 | |
| 76 | 6.48_246.1255n | 247.13273 | 6.4793333 | pos | Zedoarol | HMDB0038202 | −3.1160144 | −1.5618091 |
| 77 | 13.03_883.5013m/z | 883.50126 | 13.028717 | pos | Pitheduloside B | HMDB0034865 | −5.0778196 | |
| 78 | 14.99_377.2835m/z | 377.2835 | 14.989867 | pos | 10′-Apo-beta-caroten-10′-al | HMDB0036887 | | 35.192181 |
| 79 | 7.28_327.2176m/z | 327.21764 | 7.2776667 | neg | Corchorifatty acid F | HMDB0035919 | | 4.5032417 |
| 80 | 4.54_926.4697n | 927.47698 | 4.54215 | pos | Tragopogonsaponin B | HMDB0037911 | | 3.7664956 |
| 81 | 14.81_395.3670m/z | 395.36697 | 14.812417 | pos | Stigmasterol | HMDB0000937 | | 3.149023 |
| 82 | 6.00_280.1311n | 279.12379 | 6.00005 | neg | 13-Hydroxyabscisic acid | HMDB0036095 | | 3.0563838 |
| 83 | 7.73_329.2334m/z | 329.23337 | 7.7299667 | neg | 9,10,13-TriHOME | HMDB0004710 | | 2.4227629 |
| 84 | 5.95_197.1536m/z | 197.15357 | 5.9453333 | pos | alpha-Terpineol acetate | HMDB0032051 | | 1.9997694 |
| 85 | 5.48_280.1311n | 279.1238 | 5.4847667 | neg | Nigellic acid | HMDB0036094 | | 1.932 |
| 86 | 5.49_280.1309n | 263.12764 | 5.4853167 | pos | Crispolide | HMDB0036695 | | 1.3681446 |
| 87 | 12.19_618.3915n | 619.39874 | 12.190483 | pos | 3-O-cis-Coumaroylmaslinic acid | HMDB0034539 | | −4.4936192 |
| 88 | 8.41_644.3399n | 667.32928 | 8.40645 | pos | Goshonoside F3 | HMDB0038376 | | −35.34355 |

**Table 1.** *Cont.*

| No. | ID | *m/z* | Retention Time (min) | Ion Mode | Metabolites | Compound ID | PSP/PHP | FSP/FHP |
|---|---|---|---|---|---|---|---|---|
| Nucleosides, nucleotides, and analogues | | | | | | | | |
| 89 | 0.79_575.1100m/z | 575.10997 | 0.7941333 | neg | Orotidine | HMDB0000788 | 36.14114 | 36.162397 |
| 90 | 5.58_485.1643m/z | 485.16426 | 5.5755667 | neg | Cytidine | HMDB0000089 | 1.7393751 | |
| 91 | 0.84_244.0926m/z | 244.09257 | 0.8382333 | pos | Cytarabine | HMDB0015122 | 1.3531597 | |
| 92 | 1.19_244.0693n | 243.06197 | 1.1876833 | neg | Pseudouridine | HMDB0000767 | 1.3475125 | 2.4172629 |
| 93 | 1.98_267.0722m/z | 267.07222 | 1.9836833 | neg | Inosine | HMDB0000195 | −1.5091056 | |
| 94 | 0.81_535.0369m/z | 535.0369 | 0.81115 | neg | UDP-D-Xylose | HMDB0001018 | | 35.862012 |
| 95 | 0.82_405.0089m/z | 405.0089 | 0.8212167 | pos | Uridine 5′-diphosphate | HMDB0000295 | | 2.3748643 |
| 96 | 0.81_565.0474m/z | 565.04744 | 0.81115 | neg | Uridine diphosphate glucose | HMDB0000286 | | 1.8838349 |
| 97 | 2.16_283.0915n | 284.09878 | 2.1632167 | pos | Guanosine | HMDB0000133 | | 1.0393267 |
| Organic acids and derivatives | | | | | | | | |
| 98 | 5.59_627.2407m/z | 627.24074 | 5.5859833 | pos | 6-Hydroxysandoricin | HMDB0037556 | 1.1439601 | |
| 99 | 0.75_104.0710m/z | 104.07099 | 0.7531167 | pos | gamma-Aminobutyric acid | HMDB0000112 | 1.022634 | 1.8454808 |
| 100 | 1.12_192.0261n | 191.01882 | 1.1193833 | neg | Isocitric acid | HMDB0000193 | −0.6188259 | −1.6664215 |
| 101 | 1.13_147.0896n | 130.0863 | 1.1279 | pos | (2R,3R,4R)-2-Amino-4-hydroxy-3-methylpentanoic acid | HMDB0029449 | −1.0198727 | |
| 102 | 1.11_146.0216n | 129.0183 | 1.1108833 | pos | Oxoglutaric acid | HMDB0000208 | −1.1202847 | −2.1185923 |
| 103 | 1.11_192.0271n | 215.01603 | 1.1108833 | pos | Citric acid | HMDB0000094 | −1.1723789 | −2.4103012 |
| 104 | 0.92_324.2166m/z | 324.21664 | 0.9234167 | pos | N-Jasmonoylisoleucine | HMDB0029391 | −1.2033973 | −0.5569567 |
| 105 | 1.98_141.0182m/z | 141.01819 | 1.9789 | pos | 2-Methylene-4-oxopentanedioic acid | HMDB0037759 | −1.4134892 | −0.5977763 |
| 106 | 0.74_147.0763m/z | 147.07632 | 0.7360833 | pos | L-Glutamine | HMDB0000641 | −1.4326922 | |
| 107 | 15.27_115.0505m/z | 115.05045 | 15.2738 | pos | Ureidopropionic acid | HMDB0000026 | −1.4409771 | −0.6352039 |
| 108 | 0.55_143.0339m/z | 143.03386 | 0.5475833 | pos | 2-Methyl-4-oxopentanedioic acid | HMDB0039447 | −1.4770497 | −0.4252457 |
| 109 | 1.11_143.0339m/z | 143.03388 | 1.1108833 | pos | Oxoadipic acid | HMDB0000225 | −1.5351415 | −0.717166 |
| 110 | 4.18_202.0441m/z | 202.0441 | 4.1786667 | pos | L-Oxalylalbizziine | HMDB0039164 | −1.6272134 | −1.0362134 |
| 111 | 0.75_130.0500m/z | 130.04995 | 0.7531167 | pos | Pyroglutamic acid | HMDB0000267 | −1.6855014 | 0.2421316 |
| 112 | 0.70_175.1189m/z | 175.11886 | 0.7020333 | pos | L-Arginine | HMDB0000517 | −1.7061716 | −1.3389484 |
| 113 | 0.72_134.0447m/z | 134.04468 | 0.7190667 | pos | L-Aspartic acid | HMDB0000191 | −1.7547972 | −0.9691886 |
| 114 | 2.84_166.0862m/z | 166.08623 | 2.8391 | pos | L-Phenylalanine | HMDB0000159 | −1.8175168 | −1.0099049 |
| 115 | 0.75_119.0586n | 120.06569 | 0.7531167 | pos | L-Threonine | HMDB0000167 | −1.8452429 | |
| 116 | 0.89_118.0864m/z | 118.08643 | 0.88935 | pos | L-Valine | HMDB0000883 | −1.9042841 | −0.2565016 |
| 117 | 2.07_132.1020m/z | 132.10196 | 2.0707667 | pos | L-Isoleucine | HMDB0000172 | −1.9205672 | −1.2438406 |
| 118 | 0.84_116.0708m/z | 116.07082 | 0.8382333 | pos | 4-Amino-2-methylenebutanoic acid | HMDB0030409 | −2.1902771 | −0.7244668 |
| 119 | 0.75_132.0656m/z | 132.06558 | 0.7531167 | pos | 4-Hydroxyproline | HMDB0000725 | −2.5643691 | |
| 120 | 0.84_175.1076m/z | 175.10763 | 0.8382333 | pos | N-Acetylornithine | HMDB0003357 | −2.7493487 | |
| 121 | 1.13_307.0835n | 308.09078 | 1.1279 | pos | Glutathione | HMDB0000125 | | 37.25281 |
| 122 | 0.86_218.0902n | 219.09738 | 0.8552667 | pos | N-gamma-L-Glutamyl-D-alanine | HMDB0036301 | | 35.156087 |
| 123 | 0.77_176.1028m/z | 176.10284 | 0.7701333 | pos | Citrulline | HMDB0000904 | | 2.4286486 |
| 124 | 0.74_244.0224m/z | 244.02236 | 0.74305 | neg | O-Phosphohomoserine | HMDB0003484 | | 1.9696354 |
| Organic nitrogen compounds | | | | | | | | |
| 125 | 15.27_124.0871m/z | 124.08706 | 15.2738 | pos | L-Histidinol | HMDB0003431 | −1.4183979 | −0.6265774 |
| 126 | 15.27_122.0966m/z | 122.0966 | 15.2738 | pos | N,N-Dimethylaniline | HMDB0001020 | −1.4320285 | −0.6158732 |
| 127 | 15.29_112.0872m/z | 112.0872 | 15.291383 | pos | Histamine | HMDB0000870 | −1.4439979 | −0.6506019 |
| 128 | 12.39_300.2895m/z | 300.28947 | 12.385033 | pos | Sphingosine | HMDB0000252 | | 3.8095576 |
| 129 | 2.39_124.0395m/z | 124.03947 | 2.3868833 | pos | 2-Hydroxy-4-imino-2,5-cyclohexadienone | HMDB0031713 | | −1.8874874 |
| Organic oxygen compounds | | | | | | | | |
| 130 | 4.85_817.3868m/z | 817.38677 | 4.8532667 | neg | (3x,5x,10x)-9,10-Didehydroisohumbertiol O-[rhamnosyl-(1->4)-rhamnosyl-(1->2)-[rhamnosyl-(1->6)]-glucoside] | HMDB0040687 | 3.9229994 | |

**Table 1.** *Cont.*

| No. | ID | *m/z* | Retention Time (min) | Ion Mode | Metabolites | Compound ID | PSP/PHP | FSP/FHP |
|---|---|---|---|---|---|---|---|---|
| 131 | 10.96_369.2633m/z | 369.26334 | 10.959167 | pos | Mangalkanyl glucoside | HMDB0036015 | 3.3342658 | |
| 132 | 5.16_441.1765m/z | 441.17651 | 5.15925 | neg | Pteroside P | HMDB0036608 | 3.2783919 | |
| 133 | 10.14_676.3662n | 699.35529 | 10.144617 | pos | (S)-Nerolidol 3-O-[a-L-rhamnopyranosyl-(1->4)-a-L-rhamnopyranosyl-(1->6)-b-D-glucopyranoside] | HMDB0040846 | 3.1660281 | 3.0346403 |
| 134 | 5.16_359.1349m/z | 359.13486 | 5.15925 | neg | 2′-Methoxy-3-(2,4-dihydroxyphenyl)-1,2-propanediol 4′-glucoside | HMDB0039473 | 1.9883008 | |
| 135 | 5.45_357.1192m/z | 357.11919 | 5.4494 | neg | Moringyne | HMDB0031724 | 1.8337359 | 1.9333416 |
| 136 | 0.76_194.0418n | 193.03453 | 0.7600833 | neg | D-Glucuronic acid | HMDB0000127 | 1.7667359 | 3.7537263 |
| 137 | 0.75_356.0951n | 379.08431 | 0.7531167 | pos | 3-O-beta-D-Galactopyranuronosyl-D-galactose | HMDB0039726 | 1.7473063 | |
| 138 | 5.03_393.1767m/z | 393.17668 | 5.0349167 | neg | Foeniculoside IX | HMDB0033011 | 1.3872272 | 3.323785 |
| 139 | 5.19_463.0885m/z | 463.0885 | 5.1942167 | neg | 3′-(2″-Galloylglucosyl)-phloroacetophenone | HMDB0040622 | 1.3004351 | |
| 140 | 5.36_539.1745m/z | 539.17454 | 5.35895 | neg | Torachrysone 8-(2-apiosylglucoside) | HMDB0034612 | 1.2716972 | |
| 141 | 5.18_509.2238m/z | 509.2238 | 5.1766 | neg | Linalool 3,6-oxide primeveroside | HMDB0035489 | 1.0205958 | |
| 142 | 5.14_377.1817m/z | 377.18167 | 5.1415833 | neg | 7-Hydroxyterpineol 8-glucoside | HMDB0033019 | 0.6037439 | 1.8587819 |
| 143 | 0.76_209.0296m/z | 209.02957 | 0.7600833 | neg | Galactaric acid | HMDB0000639 | 0.4372531 | 1.7316785 |
| 144 | 5.14_355.1724m/z | 355.17236 | 5.1416 | pos | (1S,2S,4R)-1,8-Epoxy-p-menthan-2-ol glucoside | HMDB0033110 | 0.1965645 | 1.3746685 |
| 145 | 4.72_402.1525n | 447.15077 | 4.7247833 | neg | Benzyl O-[arabinofuranosyl-(1->6)-glucoside] | HMDB0041514 | −0.8390669 | −1.2836963 |
| 146 | 0.86_504.1687n | 527.15791 | 0.8552667 | pos | Gentiotriose | HMDB0029910 | −1.1018444 | −0.5817183 |
| 147 | 5.00_295.1057n | 340.10362 | 4.99905 | neg | Prunasin | HMDB0034934 | −1.1644387 | −3.8543421 |
| 148 | 9.86_329.0049m/z | 329.00487 | 9.86165 | pos | D-Sedoheptulose 7-phosphate | HMDB0001068 | −1.3360925 | −0.552754 |
| 149 | 0.79_204.0866m/z | 204.08657 | 0.7871667 | pos | N-Acetyl-D-glucosamine | HMDB0000215 | −1.3540415 | −0.6775015 |
| 150 | 0.86_522.2025m/z | 522.20253 | 0.8552667 | pos | 6-Kestose | HMDB0033673 | −1.4496564 | −0.6952694 |
| 151 | 0.87_342.1158m/z | 365.10504 | 0.8722833 | pos | Allolactose | HMDB0038489 | −1.6033747 | −0.5975339 |
| 152 | 0.85_342.1160n | 387.11424 | 0.8452333 | neg | Trehalose | HMDB0000975 | −1.6106083 | |
| 153 | 0.86_689.2101m/z | 689.21012 | 0.8552667 | pos | Mannan | HMDB0029931 | −1.6263271 | −0.6079286 |
| 154 | 0.84_288.0843n | 289.09139 | 0.8382333 | pos | Phlorin | HMDB0035589 | −1.6994496 | −0.670122 |
| 155 | 0.86_342.1158n | 360.14975 | 0.8552667 | pos | Inulobiose | HMDB0029898 | −1.7089782 | −0.7158346 |
| 156 | 14.21_589.4072m/z | 589.40716 | 14.208733 | pos | Lansioside C | HMDB0035103 | −1.8987114 | 2.4629941 |
| 157 | 0.77_144.0655m/z | 144.06547 | 0.7701333 | pos | 5-Hydroxymethyl-2-furancarboxaldehyde | HMDB0034355 | −2.2634989 | −1.3366433 |
| 158 | 0.77_164.0684n | 147.06512 | 0.7701333 | pos | 2-O-Methyl-D-xylose | HMDB0033821 | −3.5640663 | −3.2664454 |
| 159 | 4.78_469.1318m/z | 469.13181 | 4.77945 | pos | 4-Phenylbutyl glucosinolate | HMDB0038415 | | 3.9823775 |
| 160 | 0.76_383.1000m/z | 383.09996 | 0.7600833 | neg | alpha-Hydrojuglone 4-O-b-D-glucoside | HMDB0034242 | | 2.8118682 |
| 161 | 1.32_231.0838m/z | 231.08378 | 1.3224 | pos | Ethyl beta-D-glucopyranoside | HMDB0029968 | | 2.3002148 |
| 162 | 0.79_315.0933m/z | 315.09329 | 0.7941333 | neg | D-erythro-L-galacto-Nonulose | HMDB0029955 | | 1.6214985 |
| 163 | 0.81_479.1617m/z | 479.16172 | 0.81115 | neg | D-glycero-L-galacto-Octulose | HMDB0029954 | | 1.4373274 |
| 164 | 4.35_342.1311n | 365.1203 | 4.3501167 | pos | Sphalleroside A | HMDB0032767 | | −1.506083 |
| 165 | 1.13_305.0840m/z | 305.08405 | 1.1279 | pos | Arabinopyranobiose | HMDB0029619 | | −1.684668 |
| 166 | 1.13_539.1214m/z | 539.12143 | 1.1279 | pos | b-D-Glucuronopyranosyl-(1->3)-a-D-galacturonopyranosyl-(1->2)-L-rhamnose | HMDB0039728 | | −2.2164514 |

**Table 1.** *Cont.*

| No. | ID | *m/z* | Retention Time (min) | Ion Mode | Metabolites | Compound ID | PSP/PHP | FSP/FHP |
|---|---|---|---|---|---|---|---|---|
| 167 | 2.54_360.1417n | 383.13095 | 2.5369167 | pos | 2-(4-Hydroxy-3,5-dimethoxyphenyl) ethanol 4′-glucoside | HMDB0038381 | | −2.5423974 |
| 168 | 5.56_458.1789n | 503.17718 | 5.5570667 | neg | Eugenol O-[a-L-Arabinofuranosyl-(1->6)-b-D-glucopyranoside] | HMDB0037603 | | −3.5179227 |
| 169 | 4.99_295.1054n | 318.09467 | 4.9916 | pos | Sambunigrin | HMDB0034981 | | −4.9839447 |
| Organohalogen compounds | | | | | | | | |
| 170 | 13.77_226.9513m/z | 226.95127 | 13.77355 | pos | Perflutren | HMDB0014696 | −1.5064546 | −0.654059 |
| Organoheterocyclic compounds | | | | | | | | |
| 171 | 2.42_376.1367n | 399.12588 | 2.424 | pos | Riboflavin | HMDB0000244 | −1.0930927 | −2.5176871 |
| 172 | 0.77_118.0865m/z | 118.08645 | 0.7701333 | pos | 2-Methyltetrahydrofuran-3-one | HMDB0031178 | −1.231741 | |
| 173 | 15.17_175.1229m/z | 175.12292 | 15.167433 | pos | 3-(Dimethylaminomethyl) indole | HMDB0035762 | −1.4142639 | −0.6390608 |
| 174 | 15.29_147.0916m/z | 147.09158 | 15.291383 | pos | 1H-Indole-3-methanamine | HMDB0029740 | −1.425459 | −0.6368287 |
| 175 | 15.27_108.0811m/z | 108.0811 | 15.2738 | pos | 6-Acetyl-1,2,3,4-tetrahydropyridine | HMDB0030345 | −1.441196 | −0.5951493 |
| 176 | 1.79_125.0235m/z | 125.02348 | 1.7935667 | pos | 5-Hydroxymaltol | HMDB0032988 | −1.489562 | −0.594498 |
| 177 | 0.55_127.0390m/z | 127.03905 | 0.5475833 | pos | Maltol | HMDB0030776 | −1.5004781 | −0.6021647 |
| 178 | 0.86_163.0600m/z | 163.05997 | 0.8552667 | pos | D-1,5-Anhydrofructose | HMDB0041561 | −1.7541497 | −0.9841154 |
| 179 | 0.72_184.0732m/z | 184.07321 | 0.7190667 | pos | Tryptophanol | HMDB0003447 | −2.2213504 | −1.5218705 |
| 180 | 4.12_187.0633n | 188.07057 | 4.12355 | pos | Indoleacrylic acid | HMDB0000734 | −2.3607275 | −2.2746153 |
| 181 | 0.77_128.0474n | 129.05468 | 0.7701333 | pos | 3-Hydroxy-4,5-dimethyl-2(5H)-furanone | HMDB0031306 | −2.8330093 | −1.9165285 |
| 182 | 3.87_271.1150m/z | 271.11503 | 3.8704833 | pos | Neopterin | HMDB0000845 | | −1.7434786 |
| Phenylpropanoids and polyketides | | | | | | | | |
| 183 | 12.96_291.1952m/z | 291.19521 | 12.959667 | pos | Octyl 4-methoxycinnamic acid | HMDB0061861 | 8.2831723 | |
| 184 | 12.96_178.0629n | 179.07014 | 12.959667 | pos | 4-Methoxycinnamic acid | HMDB0002040 | 4.7751364 | |
| 185 | 0.76_397.0791m/z | 397.07908 | 0.7600833 | neg | Decarbamoylgonyautoxin III | HMDB0040137 | 3.5373353 | 7.5091867 |
| 186 | 14.20_379.1561m/z | 379.15614 | 14.2023 | neg | Kanzonol M | HMDB0041101 | 3.014213 | |
| 187 | 7.26_565.2866m/z | 565.28665 | 7.2588167 | neg | Hordatine A | HMDB0030461 | 2.6440336 | |
| 188 | 14.19_357.1467m/z | 357.14673 | 14.19105 | pos | [8]-Dehydrogingerdione | HMDB0039277 | 2.4931618 | |
| 189 | 0.85_695.2246m/z | 695.22456 | 0.8452333 | neg | 5-Hydroxy-7,3′,4′-trimethoxy-8-methylisoflavone 5-neohesperidoside | HMDB0030627 | 2.094353 | |
| 190 | 10.08_488.3504n | 975.69367 | 10.076433 | neg | 16beta-Hydroxystellatogenin | HMDB0040391 | 1.6017311 | |
| 191 | 5.31_624.1690n | 623.16171 | 5.305 | neg | Isorhamnetin 3-O-[b-D-glucopyranosyl-(1->2)-a-L-rhamnopyranoside] | HMDB0037085 | 1.4061339 | |
| 192 | 4.46_384.1057n | 383.09847 | 4.4620167 | neg | Eleutheroside B1 | HMDB0029549 | 1.2791289 | |
| 193 | 5.27_593.1512m/z | 593.15122 | 5.2687167 | neg | Kaempferol 3-neohesperidoside | HMDB0037573 | 1.1632915 | |
| 194 | 4.85_421.1637m/z | 421.1637 | 4.8532667 | neg | Mulberrin | HMDB0029507 | 1.0887678 | |
| 195 | 5.31_624.1684n | 625.1757 | 5.3125 | pos | Azaleatin 3-rutinoside | HMDB0037361 | 1.01554 | |
| 196 | 10.20_460.2690m/z | 460.26903 | 10.204233 | pos | Pectachol | HMDB0039064 | −1.0181638 | −0.6769391 |
| 197 | 9.44_432.2378m/z | 432.23776 | 9.4353667 | pos | Clausarinol | HMDB0041407 | −1.127634 | −0.6728349 |
| 198 | 1.30_164.0474n | 182.08123 | 1.3049 | pos | 2-Hydroxycinnamic acid | HMDB0002641 | −1.4332254 | −0.6687264 |
| 199 | 0.92_520.1013n | 543.09055 | 0.9234167 | pos | Melitric acid B | HMDB0040680 | −1.5110783 | −0.5940138 |
| 200 | 0.86_252.0633n | 253.07042 | 0.8552667 | pos | 2-O-(Z-p-Hydroxycinnamoyl)-(x)-glyceric acid | HMDB0041195 | −1.7930666 | −0.5375964 |
| 201 | 0.76_219.0449m/z | 219.04493 | 0.7600833 | neg | 3-Hydroxyflavone | HMDB0031816 | −3.0443569 | −2.8643833 |
| 202 | 0.77_418.0763m/z | 418.07634 | 0.7701333 | pos | Gonyautoxin II | HMDB0033507 | −6.0958687 | −5.450026 |
| 203 | 4.23_578.1420n | 579.14932 | 4.23295 | pos | Procyanidin B1 | HMDB0029754 | | 9.5256207 |

**Table 1.** *Cont.*

| No. | ID | *m/z* | Retention Time (min) | Ion Mode | Metabolites | Compound ID | PSP/PHP | FSP/FHP |
|---|---|---|---|---|---|---|---|---|
| 204 | 4.24_577.1352m/z | 577.13516 | 4.2357167 | neg | Procyanidin B2 | HMDB0033973 | | 8.6781467 |
| 205 | 4.16_595.1465n | 596.1538 | 4.16075 | pos | 3-Caffeoylpelargonidin 5-glucoside | HMDB0038087 | | 5.5068457 |
| 206 | 5.95_467.1864m/z | 467.18638 | 5.9453333 | pos | Thamnosin | HMDB0030550 | | 2.4912899 |
| 207 | 5.29_475.1161m/z | 475.1161 | 5.2941167 | pos | Albanin B | HMDB0034143 | | −1.0039232 |
| 208 | 5.56_571.1644m/z | 571.16438 | 5.5570667 | neg | Sakuranetin | HMDB0030090 | | −3.6529585 |

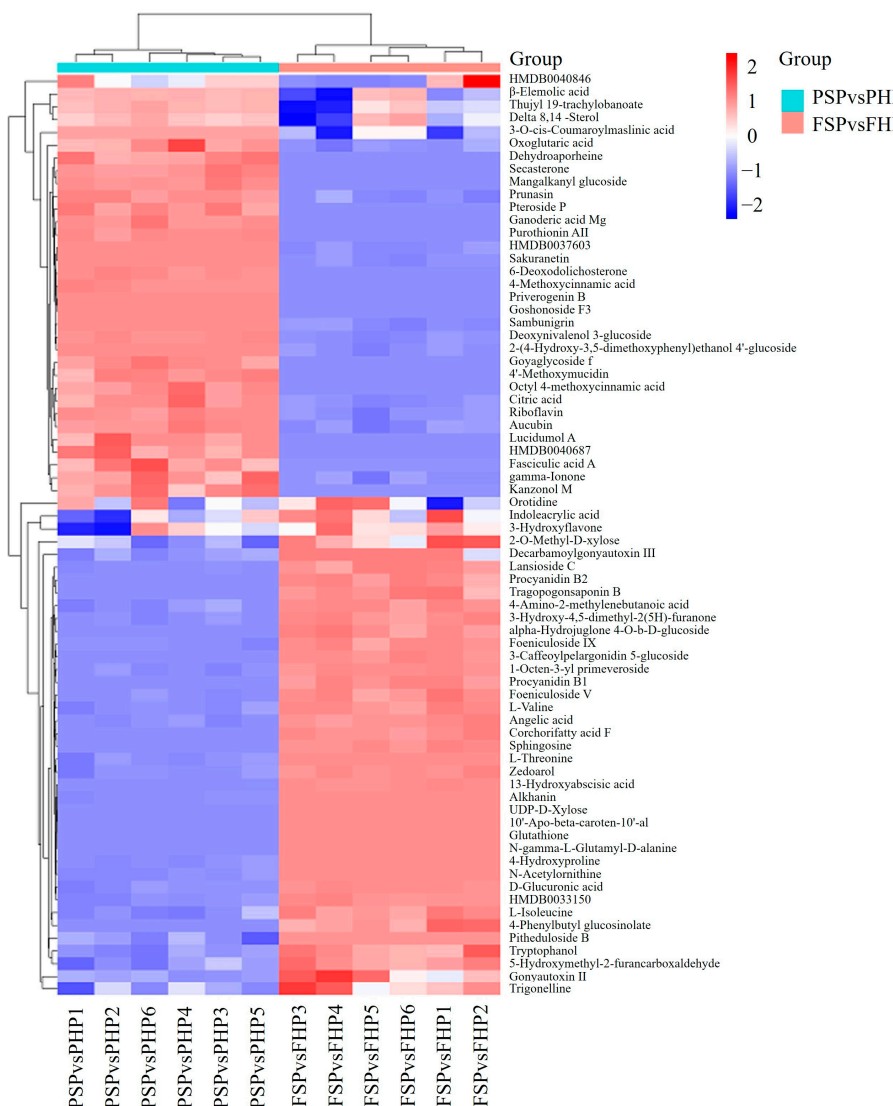

**Figure 6.** Top 40 differential metabolites in peach fruit before and after softening.

*3.5. KEGG Annotation and Metabolic Pathway Analysis*

Figure S2 shows an overview of the top 20 pathways enriched by differential metabolites in peaches before and after softening. Differential metabolite data were imported into the KEGG database to determine their position and function in related metabolic pathways. For both PSP/PHP and PTR/FHP, differential metabolites were mainly distributed in carbohydrate metabolism, amino acid metabolism, genetic information processing (aminoacyl tRNA biosynthesis and ABC transporters), and purine metabolism. In FSP/FHP, most differential metabolites were primarily involved in carbohydrate metabolism and energy production, including zeatin biosynthesis, the citrate cycle (tricarboxylic acid (TCA) cycle),

ascorbate and aldarate metabolism, pantothenate and coenzyme A (CoA) biosynthesis, nicotinate and nicotinamide metabolism, pentose and glucuronate interconversion, carbon fixation in photosynthetic organisms, glyoxylate and dicarboxylate metabolism, and amino sugar and nucleotide sugar metabolism. In PSP/PHP, most differential metabolites were mainly involved in amino acid metabolism, including arginine biosynthesis, alanine, aspartate, and glutamate metabolism, cyanoamino acid metabolism, beta-alanine metabolism, lysine biosynthesis, and arginine and proline metabolism.

## 4. Discussion

Fruit softening is the result of a series of complex physiological and biochemical reactions. Thus, a comparative investigation of flesh and peel before and after softening can clarify the mechanisms underlying variation in the ripening process. We observed a greater number of metabolites involved in analytical categories included in the KEGG databases in the groups PSP/PHP (i.e., peel) than in FSP/FHP (i.e., flesh). Nevertheless, considering the average flesh-to-skin weight ratio (25.5) and pit weight (8 g) of an individual experimental peach, the contribution of flesh by weight is over 25 times that of peel. Thus, the metabolic mechanism of peach flesh has an overall greater influence on fruit softening.

### 4.1. Degradation of Cell Wall Materials

Cell wall structural changes are generally thought to be the main factors driving fruit softening [26–28]. The distribution of cellulose is primarily observed in the primary and secondary cell walls, whereas hemicellulose forms the structural framework of the primary cell wall [29]. Furthermore, there exists a positive correlation between the contents of hemicellulose and cellulose with fruit firmness [30]. Destruction in the composition and microstructure of peach fruit cell walls during postharvest storage obviously promotes fruit softening. The cell wall hydrolases enzymatically degrade pectin, cellulose, and other polysaccharides present in the cell walls, resulting in an elevation of soluble pectin and soluble sugar content. The role of these enzymes in fruit softening has been demonstrated in various fruits such as apples [31], strawberries [32], grapes [33], and pears [34]. Our previous experiments also revealed a close relationship between polygalacturonase, β-Glucosidase, cellulase, and peach softening [11]. In this study, peach softening is accompanied by the degradation of cellulose, hemicellulose, and pectin in the cell walls of peel and flesh. We observed a significant upregulation of UDP-D-xylose and D-glucuronic acid in FSP/FHP ($|\log_2(FC)|$: 35.86 and 3.75), as well as an upregulation of UDP-glucose in FSP/FHP ($|\log_2(FC)$: 1.88). The hydrolysis of pectin produces glucuronic acid, while UDP-D-xylose is closely associated with cellulose and pectin metabolism in peaches, playing a crucial role in the metabolic pathway of amino sugars and nucleotide sugars. During this process, pectin and cellulose are degraded to form UDP-D-xylose, which is subsequently converted into UDP-glucose [35]. UDP-glucose participates in various metabolic pathways including the TCA cycle, ascorbate and aldarate metabolism, and pentose and glucuronate interconversion, thereby providing energy for storage after postharvest [35].

### 4.2. Energy Metabolism

The provision of energy is essential for the compounding and reinforcement of cell walls in plants. However, a limited supply of ATP and ADP declines the synthesis and fortification of cell walls, ultimately resulting in fruit softening [36,37]. The cellular energy status relies on the levels of ATP and ADP, with the TCA cycle and pentose phosphate pathway acting as primary suppliers for these metabolites. The metabolism of carbohydrates serves as the primary source of energy to meet the energy demands of fruit during storage, with amino sugar and nucleotide sugar metabolism representing key metabolic pathways, alongside starch and sucrose metabolism. However, after softening, there was a notable decrease in relevant metabolite levels within both TCA and pentose phosphate pathway in FSP/FHP and PSP/PHP, the content of related metabolites was significantly down-regulate, such as oxoglutaric acid, isocitric acid, citric acid, and D-sedoheptulose-7- phosphate, suggesting

an inadequate provision of cellular energy compared to pre-softening conditions. The study conducted by Zhang et al. (2023) demonstrates a strong association between the levels of ATP, ADP, and AMP as well as the activities of enzymes involved in energy metabolism with the inhibition of softening and maturity in jujubes [38]. Pearson's correlation tests were employed to analyze the relationship between energy metabolism and postharvest softening and quality decline in winter jujube fruits. The same phenomenon was observed in our experiments, wherein the softening process of peach fruit coincided with a deficiency in energy supply.

In cases where the supply of energy from carbohydrate metabolism is insufficient, there will be a significant upregulation in glycogenic amino acid and purine metabolism to compensate for the energy deficit. In this study, orotidine was significantly upregulated in both PSP/PHP ($|\log_2(FC)|$: 36.14) and FSP/FHP (36.16). The production of orotidine can be facilitated by D-sedoheptulose-7-phosphate, a metabolite derived from the pentose phosphate pathway, as well as through L-glutamine metabolism. Orotidine serves as a crucial intermediate in the de novo synthesis of pyrimidine nucleotides. When combined with phosphoribose, it forms uracil nucleotide (uridine monophosphate), which can further convert into other pyrimidine nucleotides and plays a role in monosaccharide transformation and polysaccharide synthesis. Purine metabolism, which is related to amino acid metabolism through the purine nucleotide cycle, plays crucial roles in energy supply, metabolic regulation, CoA production, and cellular growth [39,40].

The γ-aminobutyric acid was significantly upregulated in both FSP/FHP and PSP/PHP, primarily through three main metabolic pathways: alanine, aspartic acid, and glutamic acid metabolism; arginine and proline metabolism; and nicotinic acid and nicotinamide metabolism [41]. Alanine is metabolized via deamination to produce pyruvate, which enters glycolysis or the TCA cycle. Cellular L- aspartic acid is transaminated into oxaloacetic acid, as an important substrate for TCA cycle initiation and an important intermediate product of gluconeogenesis, it can also be metabolized to produce niacin, which is further converted into γ-aminobutyric acid [42]. Glutamic acid is deaminated into ketoglutaric acid, which enters the TCA cycle for ATP production and energy provision. Further metabolism of glutamic acid can produce γ-aminobutyric acid. L-arginine was significantly downregulated in both PSP/PHP ($|\log_2(FC)|$: 1.71) and FSP/FHP (1.34). In addition, citrulline was significantly upregulated, especially in FSP/FHP ($|\log_2(FC)|$: 2.43). Arginine is a polyamine that plays a crucial role in regulating cellular proliferation and differentiation while also modulating ion channels [42]. Arginine is metabolized mainly via decomposition into ornithine; the ornithine cycle generates urea, which is important for maintaining the cellular nitrogen metabolism balance [43].

In both PSP/PHP and FSP/FHP, the biosynthesis pathways of valine, leucine, and isoleucine were significantly downregulated. Specifically, valine and isoleucine were significantly downregulated in the softened peel, while isoleucine showed significant downregulation in the softened flesh. Acetohydroxy acid synthetase plays a crucial role in the biosynthesis pathways of valine, leucine, and isoleucine, as it catalyzes two molecules of pyruvate to produce one acetyl lactate and catalyzes one molecule of pyruvate and one molecule of butyric acid to form acetoxybutyric acid [44]. Acetyl lactate can further synthesize valine and leucine, whereas acetoxybutyric acid metabolism yields isoleucine as its final product. Acetohydroxy acid synthase is an enzyme encoded in the chloroplast nucleus that exhibits differential activity at different stages of plant development, but significantly decreased activity in aging tissues [45]. Downregulation of the biosynthesis of valine, leucine, and isoleucine indicates that softening of peach fruit is accompanied by its senescence.

### 4.3. Oxidative Damage

The fruit softening process is accompanied by an increase in respiratory intensity; metabolic pathways related to the respiratory chain are significantly upregulated, such as pantothenate and CoA biosynthesis, as well as nicotinate and nicotinamide metabolism.

Jiang et al. (2020) analyzed the changes in protein expression in postharvest peach fruit at different storage stages; the respiration increased, reaching a peak on day 4, at which point the fruit hardness began to show significant changes [7]. In our previous study, we detected an accumulation of reactive oxygen species (ROS), such as superoxide anion and hydrogen peroxide, during peach flesh softening [11]. The oxidative damage of cell membranes induced by ROS, which primarily occurs during respiratory metabolism, impacts fruit firmness and leads to fruit softening [46,47]. In FSP/FHP, sphingosine was significantly upregulated ($|\log_2(FC)|$: 3.81). Sphingosine is mainly derived from the degradation of sphingosine phospholipids in the cell membrane, which are important for maintaining the structure and normal function of the cell membrane [48,49]. An increase in sphingosine content in softened peaches indicates damage to the integrity of the cell membrane structure, consistent with the electron microscopy observations.

Plants can protect their cells from oxidative damage through enzymatic antioxidant defenses and non-enzymatic antioxidants [50]. Ascorbic acid-glutathione (AsA-GSH) cycle is a critical non-enzymatic antioxidant in plant cells, which removes ROS produced in the respiratory chain and maintains the cellular redox balance [50]. GSH upregulation is associated with the accumulation of superoxide anions and peroxides during fruit softening. Wang et al. (2021) showed that the oxidative damage caused by chilling injury in peaches could be reduced by regulating the ascorbic acid (AsA)–GSH cycle. Furthermore, there was a significant upregulation of glutathione (GSH) in FSP/FHP ($|\log_2(FC)|$: 37.26), primarily resulting from amino acid met down-abolism [51]. Specifically, three closely associated amino acid metabolic pathways contribute to GSH biosynthesis: alanine, aspartate, and glutamate metabolism involving the amino acids aspartate, glutamate, alanine, and γ-aminobutyric acid; arginine biosynthesis and arginine/proline metabolism encompassing the amino acids arginine, ornithine, proline, and citrulline; in addition, histidine metabolism comprising the amino acids histidine and glutamate. The metabolism of glutamate can give rise to the synthesis of glutathione. Arginine is derived from glutamic acid as a precursor, while histidine undergoes transformation via histidinase in the histidine metabolic pathway, leading to the formation of urocanic acid. Subsequently, urocanic acid is further decomposed into glutamate, which ultimately contributes to the production of glutathione.

### 4.4. Plant Hormone Regulation

Plant hormones are important factors in the regulation of soften and senescence of fruits, which have important effects on texture, flavor, and other quality during postharvest storage [52,53]. Trigonelline was significantly downregulated in both PSP/PHP ($|\log_2(FC)|$: 6.30) and FSP/FHP (3.41). Trigonelline is synthesized from nicotinic acid and is a plant hormone involved in the regulation of growth, development, and defense [53]; thus, the higher level before softening may support cell survival and growth, whereas after softening, cell growth is inhibited and its content decreases.

Abscisic acid (ABA) is considered to be an important substance in regulating soften and senescence of fruit. Studies have shown that ABA treatment can promote the expression of softening-related genes such as extensor protein, thus speeding up the ripening and softening process of strawberry fruit [54]. The oxidation pathway serves as the primary metabolic route for abscisic acid in numerous plant species. ABA undergoes oxidation to form hydroxyabscisic acid (HOABA), which is subsequently catalyzed into phaseic acid (PA) by enzymes. In most plants, PA does not accumulate and its 4′-keto groups are reduced to generate dihydrophaseic acid (DPA) or Epidihydrophaseic acid (epi-DPA). ABA levels increase in aging plant tissues along with the accumulation of its metabolites. Furthermore, research has demonstrated that under stress conditions, there is an intensified oxidation process in plants leading to an elevated rate of ABA metabolism and rapid buildup of metabolites such as DPA or epi-DPA [55,56]. In this study, 13-hydroxyabscisic acid (13-HOABA) was significantly upregulated in FSP/FHP ($|\log_2(FC)|$: 3.06), and epidihydrophaseic acid (epi-DPA) was significantly upregulated in both PSP/PHP ($|\log_2(FC)|$:

1.67) and FSP/FHP($|\log_2(\text{FC})|$: 2.52). This may be due to the accumulation of ROS that accelerates ABA oxidative metabolism. Li et al. (2023) reported that the abscisic acid content during peach soften was positively correlated with the content of most synthesis-related amino acids, suggesting a regulatory relationship between abscisic acid and amino acid metabolism [3]. In the present study, most amino acid biosynthesis pathways were downregulated, while amino acid catabolism pathways upregulated after peach fruit softening. Further studies are needed to confirm whether these changes are regulated by ABA metabolism.

Based on previous studies and our findings [3,4,7,9,11,15], we developed a model to summarize the metabolites involved in the peach fruit peel (Figure 7A) and flesh (Figure 7B) during softening.

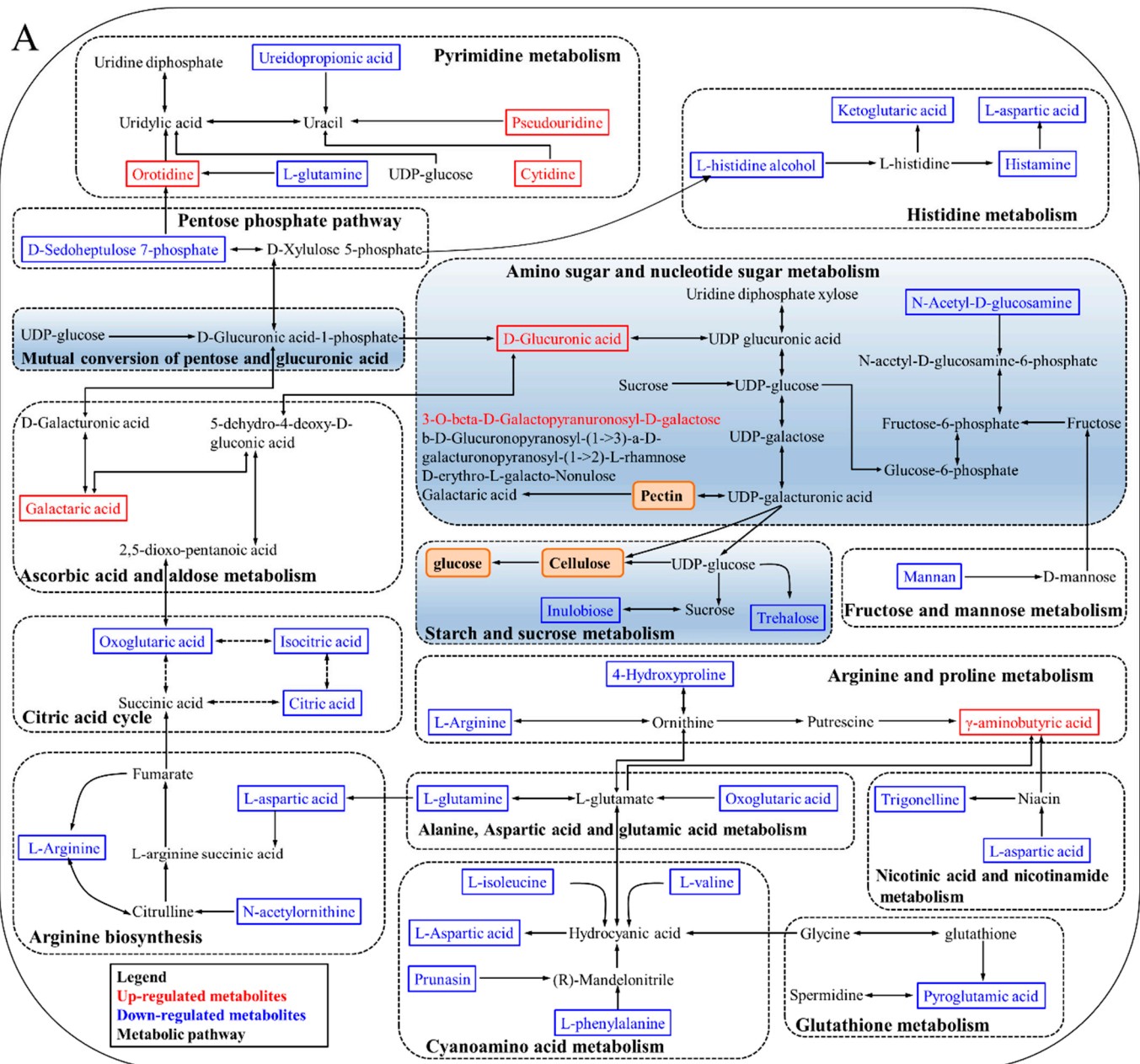

**Figure 7.** *Cont.*

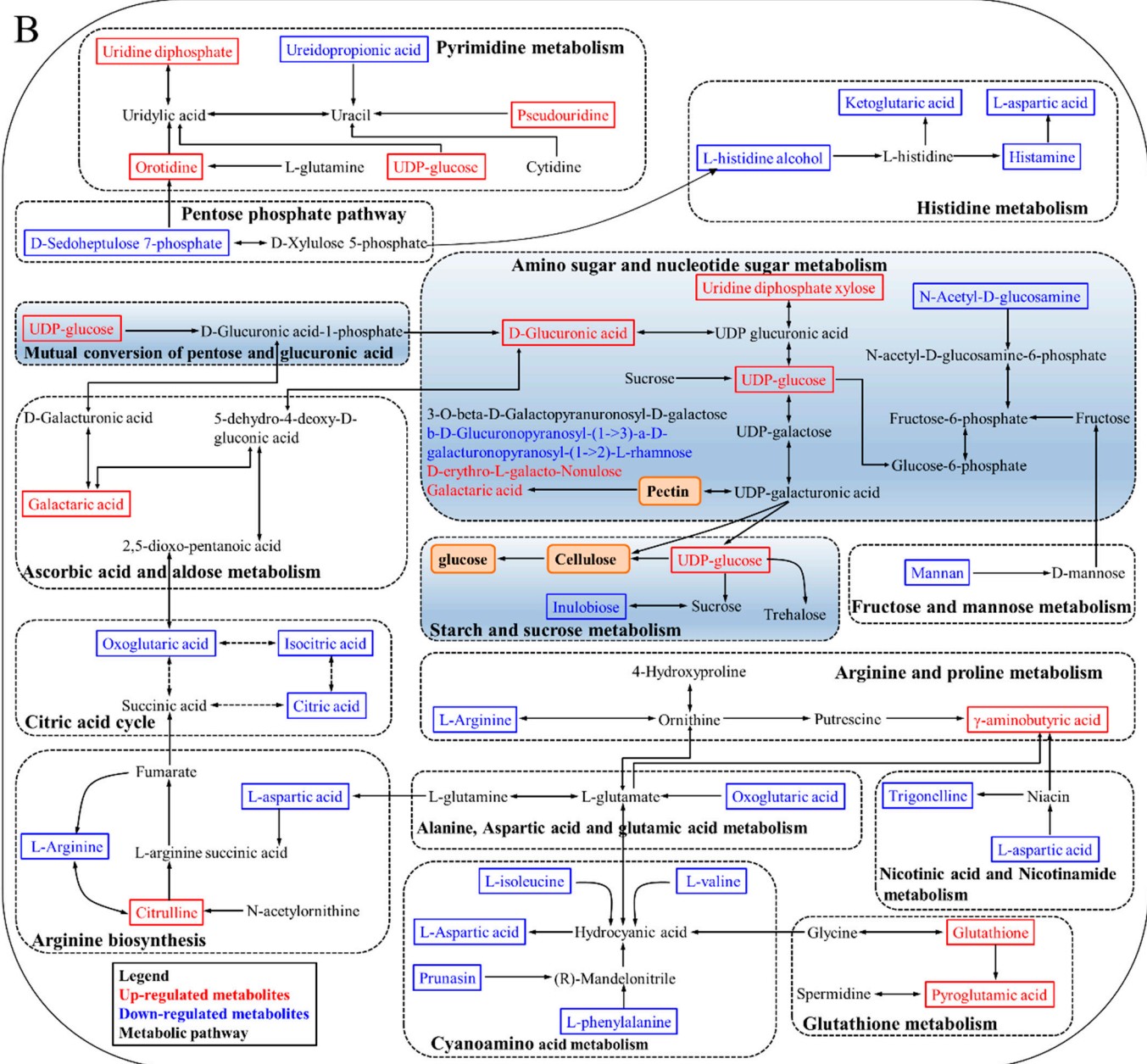

**Figure 7.** Metabolic pathways of the main metabolites in peel (**A**) and flesh (**B**) of peach fruit before and after softening. (Red indicates significantly up-regulated metabolites and blue significantly down-regulated metabolites in peach fresh after softening).

## 5. Conclusions

In this study, we investigated the mechanism of postharvest peach softening. In total, 155 and 93 significantly differential metabolites were identified from the comparative groups PSP/PHP (peel) and FSP/FHP (flesh), respectively; these metabolites included lipids, organic acids, sugars, nucleotides, phenolic acids, and flavonoids. Most were involved in carbohydrate, amino acid, purine, and energy metabolism, suggesting the involvement of these pathways in peach softening.

As a climacteric fruit, peach tissues showed a peak in respiration during storage; enhanced energy supply promoted carbohydrate metabolism, especially pectin, cellulose, and hemicellulose degradation, to provide more glycogen, and UDP-D-xylose might be one of the most key metabolites. Simultaneously, the cell walls materials were destroyed, contributing to peel and flesh softening. In cases where the supply of energy from the car-

bohydrate metabolism is insufficient, there will be a significant upregulation in glycogenic amino acid and purine metabolism to compensate for the energy deficit. The accumulation of ROS generated in the respiratory chain within cells can result in oxidative damage to cell membranes, which subsequently affects fruit firmness and leads to peach softening. At the same time, plants have the ability to safeguard their cells against oxidative damage through the utilization of antioxidants. Glutathione, a critical non-enzymatic antioxidant in plant cells, is upregulated to effectively eliminate ROS generated in the respiratory chain and maintain cellular redox homeostasis. Furthermore, plant hormones play a regulatory role in the softening process of peach fruit. Notably, the metabolism of trigonelline and abscisic acid was significantly upregulated during fruit softening.

The results of this study provide a theoretical basis for elucidating the peach softening mechanism and highlight the utility of metabolomics in mechanistic studies.

**Supplementary Materials:** The following supporting information can be downloaded at: https://www.mdpi.com/article/10.3390/horticulturae9111210/s1, Figure S1: Volcano plots of differential metabolites in peach fruit before and after softening. (A) PSP/PHP; (B) FSP/FHP; Figure S2. Top 20 KEGG pathway enriched by differential metabolites in peach fruit before and after softening. (A) PSP/PHP; (B) FSP/FHP.

**Author Contributions:** Conceptualization, X.K. and H.L. (Hong Li); methodology, investigation, Y.C., H.S., P.S. and F.Y.; data curation, H.L. (Haibo Luo); writing—original draft, X.K. and H.L. (Haibo Luo); writing—review and editing, project administration, supervision, L.Y. and H.L. (Hong Li). All authors have read and agreed to the published version of the manuscript.

**Funding:** This work was supported by the Innovation Guidance and Cultivation Project of Technological Innovation-based Enterprises (202204B1090016), Science and Technology Support for Mordern Agricultural Product Processing Technology of Yunnan Province, and Academician Expert Workstation in Yunnan Province (202005AF150007).

**Data Availability Statement:** Data are contained within the article.

**Conflicts of Interest:** The authors declare no conflict of interest.

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
