# Peer review of "Elucidating Softening Mechanism of Honey Peach (Prunus persica L.) Stored at Ambient Temperature Using Untargeted Metabolomics Based on Liquid Chromatography-Mass Spectrometry"

_horticulturae, doi:10.3390/horticulturae9111210_

Round 1
Reviewer 1 Report
Comments and Suggestions for Authors
The work presented in this manuscript falls within the scope of the journal as it offers a comprehensive Elucidation of the honey peach (Prunus persica L.) softening mechanism under storage at 25 °C using untargeted metabolomics based on liquid chromatography–mass spectrometry. The results are statistically sound and discussed in relation to the specialized literature. The discussion is clear and the results are adequately compared with previous work; it also correctly highlights the main limitations of the study, although further explanations on some points could help readers (see specific comments).
In Abstract: the sentence ''To characterize the metabolic profile before and after fruit softening, we used ultra-high-performance liquid chroma-tography with quadrupole time-of-flight mass spectrometry. Then, we identified the critical differ-ential metabolites before and after softening using multivariate statistics'' is more explaining that is not important in abstract can be explained at material and method section.
Line 1 in Introduction section, family Rosaceae has to be italic.
General comment for all indents of all subtitles has to be removed
At the 1st line of material and method the L letter has to be not italic ''Prunus persica L''.
At the results section: the Tables and figures have to be inserted within the text to be easier for readers and reviewers for understanding the text.
at the table 1, has to insert a column to the table to clear each component is detected before or after softening
In page 23, It is no need for mentioning the captions of titles as they mentioned again under each figures
Comments on the Quality of English LanguageThe MS is well written but needs to rewrite as some plagiarism is detected
Author Response
1) In Abstract: the sentence ''To characterize the metabolic profile before and after fruit softening, we used ultra-high-performance liquid chroma-tography with quadrupole time-of-flight mass spectrometry. Then, we identified the critical differential metabolites before and after softening using multivariate statistics'' is more explaining that is not important in abstract can be explained at material and method section.
Thank you for your excellent suggestion. We have deleted these two sentences.
2) Line 1 in Introduction section, family Rosaceae has to be italic.
P1, Line 31: Family Rosaceae has been modified to italic.
3) General comment for all indents of all subtitles has to be removed.
We have removed the indents of all subtitles.
4) At the 1st line of material and method the L letter has to be not italic ''Prunus persica L''.
P2, Line 92: We have revised the font format of L letter.
5) At the results section: the Tables and figures have to be inserted within the text to be easier for readers and reviewers for understanding the text.
Thank you for your kind suggestion. The Tables and figures have been inserted in the text.
6) at the table 1, has to insert a column to the table to clear each component is detected before or after softening.
Thank you for your excellent suggestion. P7, Line 225: We have added relevant information.
7) In page 23, It is no need for mentioning the captions of titles as they mentioned again under each figures.
We have deleted the captions of figures.
8) Comments on the Quality of English Language: The MS is well written but needs to rewrite as some plagiarism is detected.
Thank you for your kind suggestion. We have carefully proof read and rewrite the manuscript.
The English in this document has been checked by at least two professional editors, both native speakers of English. For a certificate, please see: http://www.textcheck.com/certificate/JSJ5Bq

Reviewer 2 Report
Comments and Suggestions for Authors
This paper is quite interesting and contributes to elucidate softening mechanism in peach fruit. However, the authors should clearly answer "What this study was performed for".

Author Response
This paper is quite interesting and contributes to elucidate softening mechanism in peach fruit. However, the authors should clearly answer "What this study was performed for".
Thank you for your kind suggestion.
P2, Line 75-81: The objective of this study was to elucidate the softening mechanism of postharvest peaches. …… Our findings clarify the mechanism underlying peach softening, and support metabolic regulation to extend their shelf life, thereby reducing peach storage and transportation losses.
General comments: …… Nevertheless, at the end of the introduction the question “What for?” has no answer, i.e.: Why is important to determine softening mechanism in this fruit? …… In my opinion, there are too many figures. Some of them could be passed to Supplementary material.
Thank you for your excellent suggestion.
P1, Line 38-42: Honey peach softening refers to the transition of the fruit from a ripe stage to an overripe stage, where moderate softening is a sign of complete maturity. Many phytochemicals are formed during the softening process [4], although excessive softening leads to postharvest quality deterioration, storage and transportation limitations, and reduced shelf life and market value.
P25-26, Line 620-626: We have removed the Figures 5 and 8 to Supplementary material.
The manuscript should have line numbers to assist in the review process.
The line numbers has added.
A list of abbreviations should help the understanding of the manuscript.
Thank you for your kind suggestion.
We have checked the full name and abbreviation throughout the manuscript, and defined the abbreviations at first time of usage.
- Introduction: The authors should specify that Peach fruit is honey peach in the first line.
We have specified that Peach fruit is honey peach.
- Materials and Methods:
The authors should specify in one sentence the 5 types of samples assayed.
Section 2.2, codes YTP, YTR, RTP, and RTR are not consistent with their corresponding words. I recommend changing them.
Thank you for your excellent suggestion. We have changed the codes.
P3, Line 97-99: The peel and flesh of hard peaches (PHP and FHP, respectively) and stored peaches (PSP and FSP, respectively) were sampled using a sharp stainless-steel knife, cut into small pieces (3–5 mm3), frozen with liquid nitrogen, and stored at −80 °C until analysis.
P3, Line 119-121: To avoid instrument errors, quality control (QC) samples were prepared by mixing all samples in equal volumes and analyzed to test the stability of the instrument system and the repeatability of sampling.
- Results
What means OPLS-DA?
P4, Line 154-155: Orthogonal partial least-squares discriminant analysis (OPLS-DA)
- Discussion
What does mean the abbreviation TCA in TCA cycle? (section 4.1 and 4.2)
P16, Line 254-255: citrate cycle (tricarboxylic acid (TCA) cycle) (section 3.5)
- Tables and Figures:
Although shown in Figure 7, the metabolites more relevant before and after softening should be marked in Table 1 and in Figure 5.
Thank you for your excellent suggestion. P7, Line 225: We have added relevant information (Table 1).
Figures 2, 7, 8, and 9 should be enlarged.
We have enlarged the Figures 2, 7, 8, and 9.
Figure A1 (code) appears in two different figures. I do not consider it necessary to show the Appendix.
We have deleted the Figure A1.
I would pass Figures 5 and 8 to Supplementary material.
We have removed the Figures 5 and 8 to Supplementary material.

Reviewer 3 Report
Comments and Suggestions for Authors
Title: Elucidation of the honey peach (Prunus persica L.) softening mechanism under storage at 25 °C using untargeted metabolomics based on liquid chromatography-mass spectrometry
Comments:
The study aimed to understand the softening mechanism of honey peach fruit during storage at 25 °C. Metabolomic analysis was conducted on the flesh and peel of honey peach using liquid chromatography-mass spectrometry. This manuscript provides a theoretical basis for elucidating the peach softening mechanism. However, there are some questions about this manuscript that need to be addressed. They are as follows:
1. More detailed information should be added to all the figure legends. For example, P23, Figure 1. The cell ultrastructure of peach fruit before (A) and after (B) softening. How did the author prepare the samples and which microscopy was used to capture the picture, what was the bar size, all this information needed to be included.
2. The study focused on the honey peach (Prunus persica L.) and may not be applicable to other peach varieties or fruits. The authors didn’t introduce why they chose honey peach.
3. The metabolomic analysis was conducted using liquid chromatography-mass spectrometry, which may have limitations in detecting certain metabolites or pathways.
4. The study did not explore the impact of postharvest treatments or interventions on peach softening, which could be relevant for practical applications in improving fruit quality during storage.
5. The study did not include functional validation of the identified metabolites or pathways, and further experiments would be needed to confirm their specific roles in peach softening.
In all, this manuscript is well-structured and has clear logic. The author employed several methods to compare the difference between the pre-harvest and post-harvest peaches to elucidate the softening mechanisms, which may play a key role in contributing to reducing peach storage and transportation losses.
Comments on the Quality of English Language
No clear grammar mistakes were found in this manuscript.
Author Response
- More detailed information should be added to all the figure legends. For example, P23, Figure 1. The cell ultrastructure of peach fruit before (A) and after (B) softening. How did the author prepare the samples and which microscopy was used to capture the picture, what was the bar size, all this information needed to be included.
Thank you for your excellent suggestion. We have added the detailed information.
P3, Line 102-110: The cell ultrastructure of peach peel and flesh were visualized as previously described by Luo et al. (2019), with some modifications [24]. Tissue blocks of approximately 1 mm³ were sliced from peach surface and washed three times with cold phosphate-buffered saline (PBS, pH7.0, 0.1 M) for 15 min each. The samples were soaked in 2.5% (w/v) glutaraldehyde for 24 h at 4 °C, washed with PBS three times, and then soaked in 1% osmic acid fixative solution for 2 h. The samples were washed with PBS (pH7.4) three times, and dehydrated in 50%, 70% and 90% ethanol for 15 min each, followed by 100% ethanol for 20 min. After fixing with conductive carbon adhesive and spraying gold with an ion sputtering instrument for 50 s, and the samples were observed under a FEI Nova Nano 450 scanning electron microscope (FEI Company, USA).
- The study focused on the honey peach (Prunus persica L.) and may not be applicable to other peach varieties or fruits. The authors didn’t introduce why they chose honey peach.
Thank you for your kind suggestion.
P1, Line 36-42: However, honey peaches are climacteric fruits with a vigorous postharvest respiratory physiological metabolism. Honey peach softening refers to the transition of the fruit from a ripe stage to an overripe stage, where moderate softening is a sign of complete maturity. Many phytochemicals are formed during the softening process [4], although excessive softening leads to postharvest quality deterioration, storage and transportation limitations, and reduced shelf life and market value.
- The metabolomic analysis was conducted using liquid chromatography-mass spectrometry, which may have limitations in detecting certain metabolites or pathways.
Thank you for your excellent suggestion.
P2, Line 64-74: Untargeted metabolomics can rapidly identify and classify metabolites based on differences in metabolic pathway maps, and based on LC-MS/MS, can reliably analyze metabolic profiles, although it have limitations in detecting certain metabolites or pathways.
- The study did not explore the impact of postharvest treatments or interventions on peach softening, which could be relevant for practical applications in improving fruit quality during storage.
Thank you for your kind suggestion.
In our previous study, Feicheng honey peach fruit was used as a test material to investigate the synergistic preservation effect of 1-methylcyclopropene (1-MCP) and laser microporous film (LMF) packaging. The co-application of 1-MCP and LMF significantly inhibit the respiration, reduce the degradation of cell wall materials, and delay ROS accumulation and membrane lipid peroxidation in honey peach fruits during cold storage; these changes can slow the softening of peach fruit.
[11] Li, X.; Peng, S.; Yu, R.; Li, P.; Zhou, C.; Qu, Y.; Li, H.; Luo, H.; Yu, L. Co-application of 1-MCP and laser microporous plastic bag packaging maintains postharvest quality and extends the shelf-life of honey peach fruit. Foods 2022, 11, 1733. http://dx.doi.org/10.3390/FOODS11121733
P2, Line 75-76: The objective of this study was to elucidate the softening mechanism of postharvest peaches.
P16, Line 282-284: Our previous experiments also revealed a close relationship between polygalacturonase, β-Glucosidase, cellulase, and peach softening [11].
P16, Line 360-362: In our previous study, we detected an accumulation of reactive oxygen species (ROS), such as superoxide anion and hydrogen peroxide, during peach flesh softening [11].
- The study did not include functional validation of the identified metabolites or pathways, and further experiments would be needed to confirm their specific roles in peach softening.
Thank you for your kind suggestion. The functional validation of the identified metabolites and pathways will be conducted in our further study.
